# Genetic determinants of monocyte splicing are enriched for disease susceptibility loci

Isar Nassiri[1,2,3,4,5] ✉, James J. Gilchrist [6,7], Orion Tong [6], Evelyn Lau[2], Sara Danielli[5], Hussein Al Mossawi [8], Matthew J. Neville [9,10], Julian C. Knight [2,11] & Benjamin P. Fairfax [5,6] ✉

Insights into variation in monocyte context-specific splicing and transcript usage are limited. Here, we perform paired gene and transcript QTL mapping across distinct immune states using RNA sequencing data of monocytes isolated from a cohort of 185 healthy Europeans incubated alone or in the presence of interferon gamma (IFN-γ) or lipopolysaccharide (LPS). We identify regulatory variants for 5749 genes and 8727 transcripts, with 291 context-specific transcript QTL colocalizing with GWAS loci. Notable disease relevant associations include IFN-γ specific transcript QTL at COVID-19 severity locus rs10735079, where allelic variation modulates context-specific splicing of *OAS1*, and at rs4072037, a risk allele for gastro-esophageal cancer, which associates with context-specific splicing of *MUC1*. We use DNA methylation data from the same cells to demonstrate overlap between methylation QTL and causal context-specific expression QTL, permitting inference of the direction of effect. Finally, we identify a subset of expression QTL that uncouple genes from proximally acting regulatory networks, creating 'co-expression QTL' with different allele-specific correlation networks. Our findings highlight the interplay between context and genetics in the regulation of the monocyte gene expression and splicing, revealing putative mechanisms of diverse disease risk alleles including for COVID-19 and cancer.

Dysfunctional innate immunity plays a role in the pathogenesis of diverse human disease processes, with chronic inflammation implicated in autoimmunity and cancer[1], whilst impaired acute immune responses can contribute to susceptibility to infection[2]. Circulating monocytes play a central role in the early innate immune response, and inter-individual variation in monocyte gene expression results in variation in functional activity. Monocytes display stereotypical responses to different immune stimuli, with their activation leading to widespread changes in gene expression, with parallel changes in chromatin accessibility, revealing context-specific regulatory variants. Consequently, mapping monocyte expression quantitative trait loci (eQTL) across different activation states provides mechanistic insights into divergent innate immunity with relevance to disease processes[3,4]. Whilst early context-specific eQTL analyses were based

[1]Oxford-GSK Institute of Molecular and Computational Medicine (IMCM), Centre for Human Genetics, Nuffield Department of Medicine, University of Oxford, Oxford, UK. [2]Centre for Human Genetics, Nuffield Department of Medicine, University of Oxford, Oxford, UK. [3]Nuffield Department of Clinical Neurosciences, University of Oxford, Oxford, UK. [4]Department of Psychiatry, University of Oxford, Oxford, UK. [5]Department of Oncology, University of Oxford, Old Road Campus Research Building, Oxford, UK. [6]MRC Weatherall Institute of Molecular Medicine, University of Oxford, Headington, Oxford, UK. [7]Department of Paediatrics, University of Oxford, Oxford, UK. [8]Nuffield Department of Orthopaedics Rheumatology and Musculoskeletal Sciences, University of Oxford, Oxford, UK. [9]Oxford Centre for Diabetes, Endocrinology and Metabolism, University of Oxford, Oxford, UK. [10]NIHR Oxford Biomedical Research Centre, OUH Foundation Trust, Oxford, UK. [11]Chinese Academy of Medical Science Oxford Institute, University of Oxford, Oxford, UK. ✉e-mail: isar.nassiri@ndcn.ox.ac.uk; Benjamin.fairfax@oncology.ox.ac.uk

on microarray-based approaches[3,5,6], these have been greatly complemented by more recent single-cell RNAseq studies[7]. However, neither of these approaches provides a comprehensive perspective on activation induced transcript modulation, including splicing and differential transcript usage. To further our understanding of the impact of genetic variants on splicing and differential transcript usage, we have conducted eQTL mapping using paired-end RNA-seq at gene (gQTL) and transcript (tQTL) levels, integrating observations with methylation QTL (mQTL). We study how monocyte genetics and context-specific transcriptomics relate in different immune contexts (Fig. 1).

We find that, consistent with gQTLs, tQTLs show a high degree of context specificity, with 54.6% (4763/8727) observed in one condition only (FDR < 0.01). By integrating our results with genetic associations within the UK Biobank, we explore the potential contribution of context-specific tQTLs to human disease processes[8]. Notably, we find that 6.1% (291/4763) of context-specific tQTLs associate with GWAS disease risk loci[9], including those linked to severe COVID-19[10]. The connection between genetic variation, DNA methylation, and gene expression is intricate, with methylation quantitative trait loci (mQTLs) frequently underpinning differential gene expression[11]. Whilst IFN-γ has minimal effects on monocyte DNA methylation during the timeframe we evaluated, LPS causes highly specific and punctate effects[12]. To investigate the relationship between context-specific g/tQTL and mQTL, we integrated DNA methylation status with gene expression from untreated and LPS-treated monocytes[13,14], finding 89.9% of post-LPS g/tQTL:mQTL pairs shared a causal variant. Finally, we demonstrate that for a subset of gQTL and tQTL, the regulatory variant alters the relationship between the *cis* gene and co-regulated gene networks, leading to the formation of co-expression QTL (coExQTL)[15,16], which we describe and replicate across different activation states. We propose coExQTLs provide paradigmatic insights into the mechanisms whereby small-scale regulatory variation may induce large-scale impacts on phenotype. Our work further highlights the need to consider context when determining the effect of regulatory variant function and provides insights into genetic determinants of transcriptional pathway activity.

## Results

### Identification of cis-acting QTL

We performed genome-wide gQTL and tQTL analysis to identify loci associated with both gene expression and transcript usage in monocytes incubated for 24 h in media alone or in the presence of either LPS or IFN-γ (Supplementary Fig. 1). We filtered out technical variability from gene read counts and included an optimal number of principal component covariates in the g/tQTL analysis[17–19]. For gQTL analysis, mapping was performed using a 1 Mb window centred on the transcription start site (TSS) of the gene of interest to capture distal enhancers and long-range acting variants, whereas for tQTL, a 100Kb window was considered, given the known more local regulation of splicing and to minimise multiple testing[20–27]. Our stepwise conditional analysis revealed independent g/tQTL[28], implying multiple significant independent associations for a subset of genes after conditioning for the lead variant associated with it. We identified a total of 26,500 independent gQTL (± 1 Mb, FDR < 0.01) across 10875 genes and 13822 independent tQTL (± 100 kb, FDR < 0.01) involving 5749 genes and 8727 transcripts. Of these, we identified 3441 gQTL and 4763 tQTL in one condition only, whilst 1937 gQTL and 2646 tQTL were observed only in the stimulated state (significant in IFN-γ and/or LPS treated conditions but not in the naïve state) (Fig. 2a–c) (Supplementary Data 1).

The Moloc method was utilised to further evaluate the context specificity of g/tQTL identified in RNA-seq data[29,30], enabling comparison of evidence for shared or independent effects of genetic variants whilst mitigating the impact of linkage disequilibrium. This approach identified 3572 gQTL and 2016 tQTL with evidence of specificity to one condition only (PP > 0.5, Supplementary Data 2). The median distance for gQTL from the transcription start site (TSS) was 43,649 bp (95% CI: 42,488 – 44,449 bp), whilst tQTL are typically more proximal to TSS (median 19,481 bp 95% CI: 18,233 – 20,455 bp (UT), 19,285 bp 95% CI: 18,123 – 20,397 bp (LPS), 18,263 bp 95% CI: 17,257 – 19,433 bp (IFN-γ)). The differences in window size used to identify gQTL and tQTL complicate accurate comparison of distances between these types of regulatory loci and the TSS. However, when we focused on independent gQTL within 100 kb windows, we observed that the distribution of distances from TSS to gQTLs and

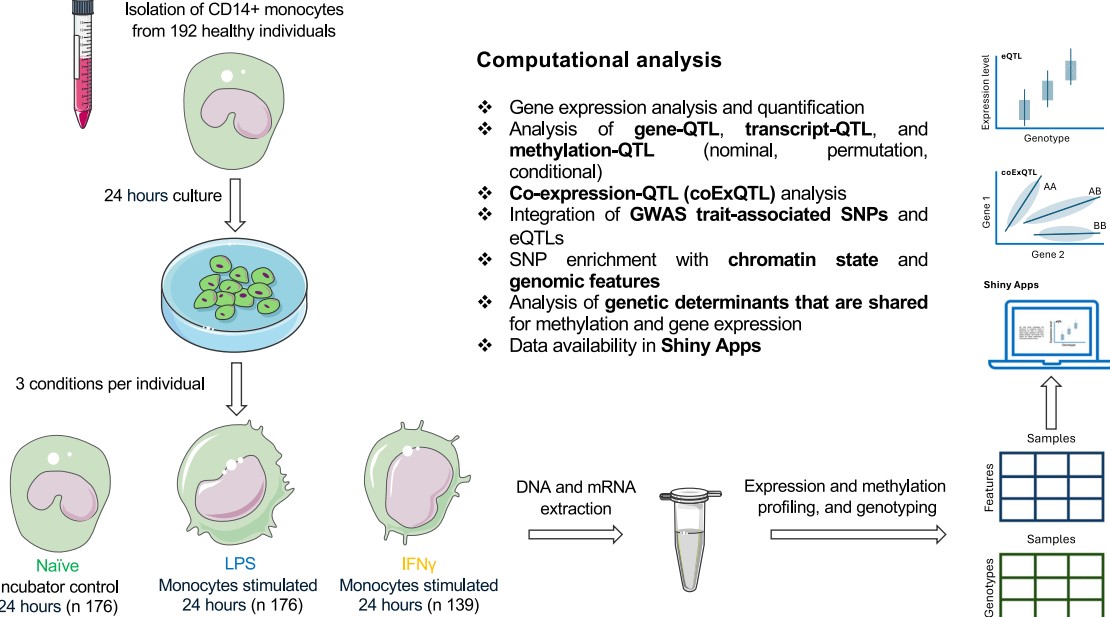

**Fig. 1 | Overview of approach.** Paired gene and transcript QTL mapping, using paired-end RNA-seq of primary monocytes that had been incubated in an untreated state or following exposure to IFN-γ or LPS. Imputed genotypes were derived from array-based genotyping. g/t/mQTL mapping was performed using QTLtools with biological significance and potential implications for disease inferred via integration with GWAS trait-associated SNPs. All data is available via a web browser.

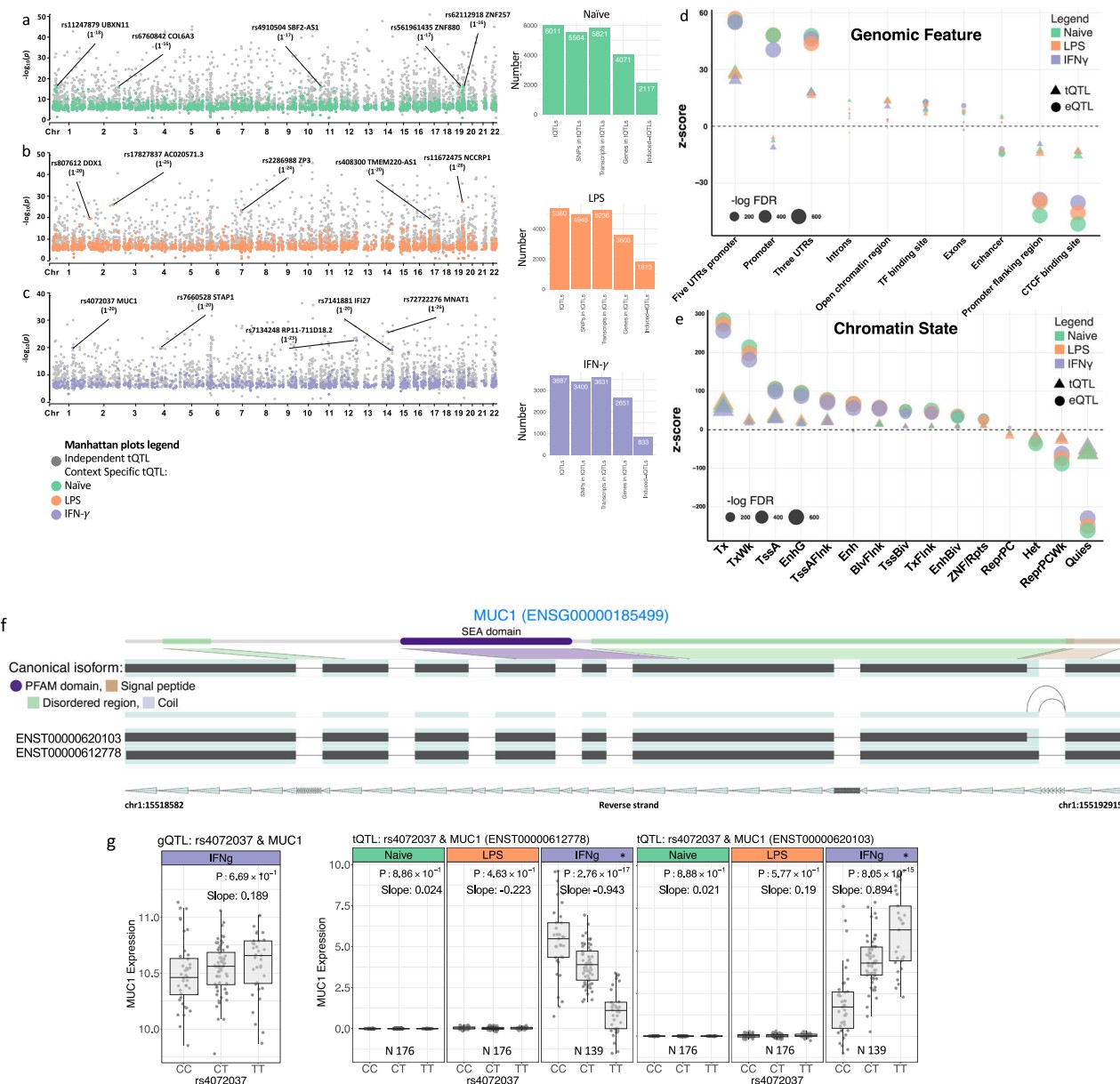

**Fig. 2 | SNP and gene mapping in the tQTL profiles.** Manhattan plots of independent monocyte tQTL-associations for the naïve state (**a**), or after treatment with LPS (**b**), or IFN-γ (**c**). Top condition-specific independent SNPs can be found in highlighted SNPs on Manhattan plots. The *y*-axis displays the -log$_{10}$(FDR) of each association, while the x-axis displays the chromosomal position of SNPs. **d** Functional annotation of independent g/tSNPs using genomic features. The plot displays the z-score calculated using the proportion of SNPs (independent g/tSNP and their LD proxies) which have corresponding functional annotations assigned by biomaRt (Ensembl) and UCSC databases in comparison with background SNP. The foreground SNP sets for each gene/transcript were made up of g/tSNP with FDR <0.001, and other gSNPs within the 1 Mb and tSNPs within the 100 Kb window around the TSS were used as background. **e** The enrichment of independent g/tSNP with chromatin state (E029 chromatin states dataset provided by the Roadmap Epigenomics Consortium) across primary monocytes from peripheral blood tissue. **f** Highlighted isoforms (red colour) of *MUC1*, including *ENST00000612778* and

*ENST00000620103* show significant tQTL in IFN-γ stimulated condition. **g** For *MUC1*, we found an independent SNP (rs4072037) which is associated with two transcripts in a context-specific manner. The display of splice junctions is limited to those that are not present in all transcripts. Abbreviations: *TssAFlnk* Active TSS Flank, *TxFlnk* Transcription, *Tx* Transcription Flank, *TxWk* Weak Transcription, *EnhG* Gene Enhancer, *Enh* Enhancer, *ZNF/Rpts* Zinc Finger/Repeats, *Het* Heterochromatin, *TssBiv* Bivalent TSS, *BivFlnk* Bivalent Flank, *EnhBiv* Bivalent Enhancer, *ReprPC* Repressed Polycomb-repressed Complex, *ReprPCWk* Weak Repressed Polycomb-repressed Complex, *Quies* Quiescent chromatin. *fiveUTRs* 5′ Untranslated Regions. *threeUTRs* 3′ Untranslated Regions. "N" represents and lists the specific number of donors. The x-axis of the boxplot represents the different alleles of the SNP that are related to the expression. Each box indicates a particular SNP allele, and the *y*-axis displays the expression levels of individuals with that allele. The normalised read count and Fragments Per Kilobase Million (FPKM) metrics display the expression levels of genes and transcripts, respectively.

tQTLs with the 100 Kb window size remained significantly different (Welch Two Sample *t* test *P* < 0.01), which suggests divergent regulatory mechanisms (Supplementary Fig. 2). In keeping with this, gQTL and tQTL exhibited distinct enrichment for certain genomic features, with promoter enrichment observed only for gQTL, whereas

tQTL demonstrated relative enrichment within enhancers and regions of open chromatin (Fig. 2d).

We tested g/tSNPs at gQTL and tQTL peaks for enrichment of chromatin state features for primary human monocytes[31]. We found enrichment of tSNPs at gene 5′ and 3′ transcription flanking (TxFlnk)

marks, whereas gSNPs were enriched at active transcription start sites (TssA), with both depleted in quiescent/low epigenetic signals (Fig. 2e).

Replication of gQTL is important in validating observed genetic determinants of expression and is particularly useful for well-characterised context-specific observations[20]. To evaluate the reproducibility of the cis-gQTLs identified using RNA-seq in this cohort, we compared results with those from microarray analysis of an earlier cohort treated in the same manner[3]. We considered gQTLs reported in the microarray dataset based on commonality of gene symbols with the RNA-seq-based data, comparing SNP-gene pairs, and their gQTL p-values and effect sizes.

This demonstrated 7675/10139 total (75.7%) naïve, 7344/9831 (74.7%) total LPS, and 7293/9410 total (77.5%) IFN-γ significant independent gQTL at FDR < 0.01 that were previously reported in the gQTL Catalogue[32]. Notably, 95.6% (naïve), 94.2% (LPS), and 95.2% (IFN-γ) of these replicated gQTL corresponded to the previously reported independent SNP[32], with 99% demonstrating identical allelic directions (correlation coefficient = 0.98, Shapiro-Wilk normality test p-value < 0.001, Supplementary Fig. 3).

To evaluate whether identified gQTL could be replicated at the single-cell level, we compared gQTL results with published single-cell RNA-seq (scRNA-seq) data of monocytes from 120 PBMC samples either stimulated or exposed in vitro to *Pseudomonas aeruginosa* (PA24)[7]. The scRNA-seq dataset was analysed for gQTLs based on shared gene symbol, comparing SNP-gene pairs and their gQTL p-values. This demonstrated general consistency of results with 41,769/91,678 (45.5%) representing naïve and 47,931/95,226 (50.3%) representing total PA24 replication. The datasets showed a significant positive correlation between the effect sizes of gQTLs (naïve 95.6% and PA24 96.6%) (Supplementary Data 3), indicating that the associations identified can be reproduced at a single-cell level.

We found that 54.6% of tQTL (4763 observed over 8727 total) showed context specificity, of which 6.5% (568 observed over 8727 total) involve more than one transcript associating with the same regulatory variant (387/568 demonstrating context-specificity). Whilst 39.3% (4276 observed over 10875 total) of gQTL genes had a tQTL, with 16.6% (710 observed over 4276 total) of them being context-specific gQTL genes, notably 25.6% (1473 out of 5749) of tQTL were to genes without gQTLs. Thus, tQTL analysis provides additional information regarding context-specific regulatory variant activity that complements gQTL analysis.

An illustrative example of the context-specific splicing occurring across conditions is at the gene *MUC1*. Alternative transcripts of *MUC1*, including *ENST00000612778* (FDR 2.7 × 10⁻¹⁷) and *ENST00000620103* (FDR 8.0 × 10⁻¹⁵), showed tQTL (but not gQTL) upon exposure to IFN-γ (Fig. 2f), mapping to rs4072037, a risk allele (rs4072037:G) for oesophageal and gastric cancer, highlighting the potential for insights into mechanisms of disease of such features (Fig. 2g)[33,34]. Multiple further examples of tQTLs demonstrating high conditional specificity and with opposing direction of effect for different transcripts were noted at genes including *RGS1*, *DDX1*, *CTSC*, and *KIFC3* (Supplementary Figs. 4–6).

## Disease association

To formally explore disease and trait associations with identified monocyte g/tQTL across contexts, we integrated observations with the UK Biobank and GWAS summary statistics for 380 traits. We employed Mendelian randomisation (MR) to infer causal relationships between exposures (gene expression) and outcomes (traits). We found that both gQTLs and tQTLs identified across contexts in this analysis were enriched for disease-associated GWAS loci (FDR < 5 × 10⁻⁸ and PPH4 > 0.8).

A total 126 trait-associated gQTL were observed, whereas 291 tSNPs were found to colocalise with 140 traits. Whilst 95 traits were associated with both tQTL and gQTL, a further 45 traits were found to

colocalise with tSNPs, demonstrating the additional potential disease insights from such analysis. In keeping with differential transcript usage across contexts playing a key role in immune-mediated disease susceptibility, enrichment of condition-specific tQTLs with GWAS risk alleles was greater than that for gQTLs (Fig. 3a, b).

Although we used different window sizes for identifying gQTL and tQTL, secondary analysis limited to the 100 Kb window size for independent gQTLs confirmed that these distinct observations were not due to divergent window size usage, with 90.9% GWAS trait colocalization amenable to replication (UT: 100%, LPS: 100%, IFN-γ: 63.2%) (Supplementary Data 4).

This analysis demonstrated untreated monocytes exhibit significant causal relationships in tQTL for rheumatoid arthritis and cancer, whereas stimulated monocytes demonstrated the most causal relationships for asthma (Fig. 3b and Supplementary Data 4), for which LPS-induced cytokines and IFN-γ induced dendritic cell differentiation play key roles[35]. These data provide further granularity into potential GWAS mechanisms, demonstrating that variants at disease-risk loci are associated with the use of activated monocyte isoforms[32,36–38].

Both IFN-γ and LPS elicit prominent type I interferon induction, a key characteristic of early anti-viral innate immune responses, including those to SARS-CoV-2. Exemplifying the relevance of stimulated monocytes to the genetics of COVID-19 pathogenesis, we observed several context-specific gQTL and tQTL colocalising with COVID-19 severity risk loci (Fig. 3c–e). A leading example of these is within the antiviral restriction enzyme activators (*OAS*) gene cluster, where gQTL to *OAS1* and *OAS3* colocalise with the COVID risk locus rs10735079[10]. Whereas in untreated monocytes this locus displays weak gQTL activity (*OAS1* FDR = 0.0027, *OAS3* FDR = 0.0023), the expression of these genes is robustly induced by IFN-γ, leading to a markedly increased significance of gQTL associations (Fig. 3d). Analysis of transcript usage at this locus demonstrates complex transcript switching, specifically in *OAS1* isoform usage (involving *ENST00000202917*, *ENST00000452357*), most significantly post-IFN-γ (Fig. 3e and Supplementary Data 5). Differential splicing at rs10735079 (chr12:112942203 G > A) leads to exon skipping between *ENST00000452357* and *ENST00000202917,* with this SNP forming the most likely causal variant for both post-IFN-γ gQTL, tQTL and severe COVID-19 susceptibility (PPH4 = 0.99) (Fig. 3c–e), demonstrating multifaceted regulatory activity at this locus in a disease-relevant state. A further key COVID-19 GWAS locus demonstrating context-specific activity is rs6517156, which forms a gQTL for *IFNAR2* post IFN-γ (FDR 6.2 × 10⁻⁶), again emphasising the importance of the IFN-γ stimulus to elucidation of the COVID-19 disease state (Fig. 3f). Our findings are in accordance with previously published results and provide further resolution of the effect of this COVID severity locus at the transcript level[39,40].

## Shared genetic determinants on methylation and gene expression

To explore the relationship between context-specific g/tQTL formation and variation in DNA methylation, we assessed DNA methylation from the same samples in the untreated state and post LPS[11,41,42]. We performed genome-wide methylation quantitative-trait loci (mQTL) analysis to identify variants associated with DNA methylation levels, identifying a total of 19,962 mQTL (17,279 untreated, 16,853 post LPS; FDR < 0.01, Supplementary Data 1).

Colocalization analysis was used to identify g/tQTL and mQTL pairs likely sharing a causal variant[43]. For mQTLs with multiple associated CpGs, we selected the CpG with the most significant FDR and the highest posterior probability of a causal hypothesis. We found one causal variant to be associated with both expression and methylation (posterior probabilities of PPH4 > 0.8, FDR < 0.01)[9] for 45.7% (497/1086) of gQTL-mQTL pairs in naïve monocytes and 46.3% (365/787) post LPS (Fig. 4a, Supplementary Data 6). Context-specific tQTLs that

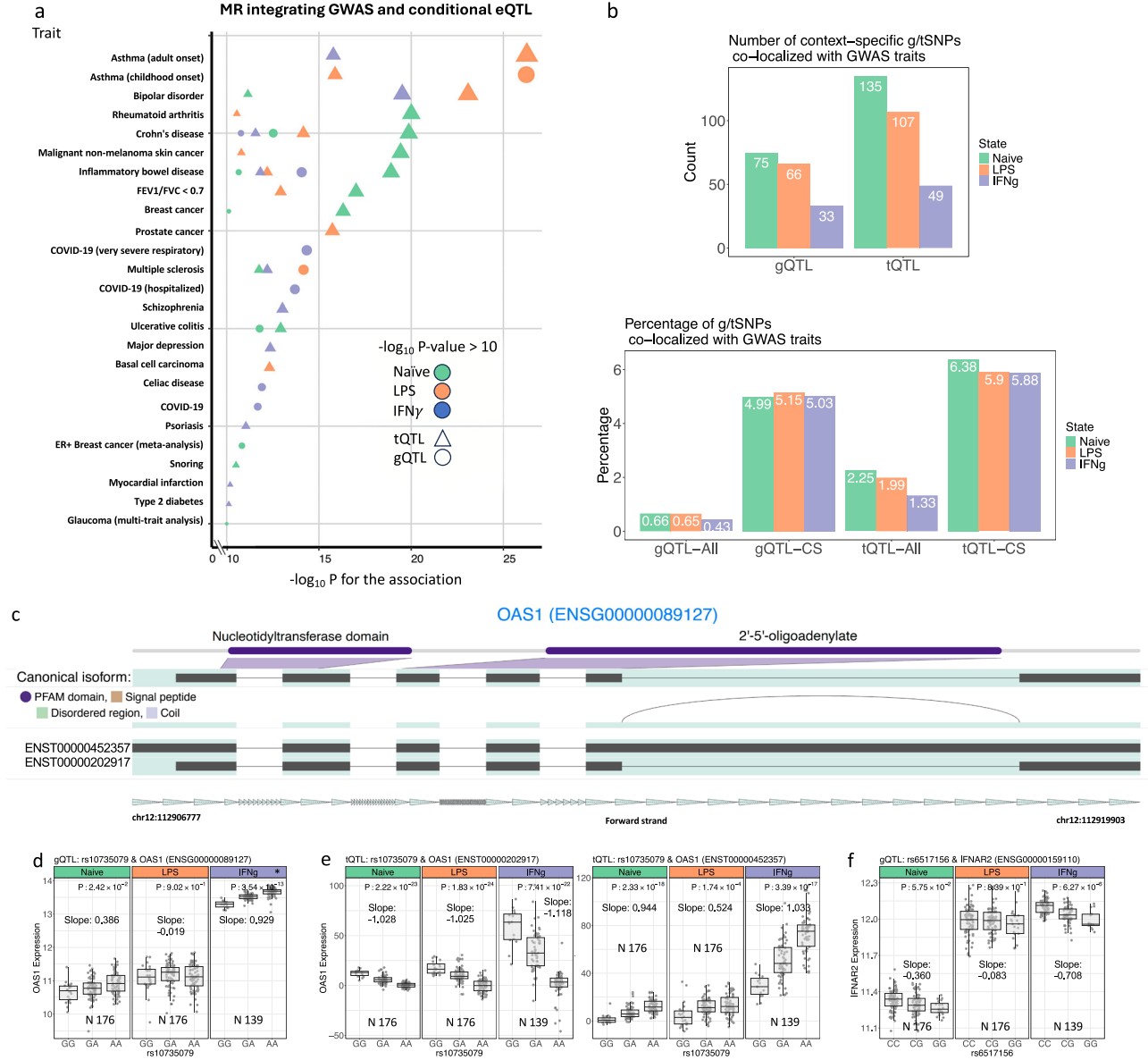

**Fig. 3 | Disease associations of g/tQTL. a** Context-specific g/tQTL colocalising with GWAS disease risk loci (PPH4 > 0.8 and GWAS p-value threshold of $5 \times 10^{-8}$). **b** Most significant trait association of independent g/tSNP using GWAS summary statistics and Mendelian randomisation (posterior probability PPH4 > 0.8 and FDR < $5 \times 10^{-9}$). The results represent a significant causal relationship between traits related to autoimmunity and inflammation, with enrichments where $P < 1 \times 10^{-10}$ plotted. **c** Highlighted isoforms of *OAS1,* including *ENST00000202917* and *ENST00000452357,* show a significant switch in isoform usage across untreated and stimulated conditions. ENST00000452357 and ENST00000202917 are generated through *OAS1* SNP-dependent splicing. In ENST00000452357, there is no use of a splice donor in exon 5. Exons 5 and 7 have been combined in ENST00000202917. The display of splice junctions is limited to those that are not present in all transcripts. **d** rs10735079, a severe COVID-19 risk locus, influences the gene level expression of *OAS1.* **e** rs10735079 demonstrates divergent tQTL effects to *OAS1* transcripts *ENST00000202917* and *ENST00000452357* most significantly post IFN-γ. **f** gQTL to COVID-19 risk locus at *IFNAR2* enhanced post IFN-γ. Each data point on the horizontal and vertical axis indicates values for a single sample. Regression lines are shown for categories of genotypes. Abbreviations: "N" represents and lists the specific number of donors. The x-axis of the boxplot represents the different alleles of the SNP that are related to the expression. Each box indicates a particular SNP allele, and the y-axis displays the expression levels of individuals with that allele. The normalised read count and Fragments Per Kilobase Million (FPKM) metrics display the expression levels of genes and transcripts, respectively. The 25th and 75th percentiles are represented in the box at the bottom and top. The box has a line that indicates the median of the expression. The whiskers cover both the minimum and maximum values, with the exception of outliers.

share the same independent and causal variant with mQTLs were identified in naïve and post-LPS by 57.4% (862/1501) and 53.2% (590/1107), respectively (Fig. 4a and Supplementary Data 6).

Where there was a relationship between CpG methylation for mQTL and gene expression, we tested the directional causality of these loci in terms of genetic variation, DNA methylation, and gene expression by applying Steiger directionality tests across g/tQTL-mQTL pairs.

In the naïve state, 36% (179/497) of gQTL and 42.4% (366/862) of tQTL demonstrate dependent causal effects on methylation linked to expression or vice versa, with this proportion being 40.3% (147/365) of gQTL and 43.3% (256/590) of tQTL post-LPS. Steiger directionality tests were used to determine the direction of the regulatory relationship for the identified pairs of tQTLs and mQTLs that share a likely causal variant. The analysis revealed that in 41% of naïve and 41.9% of post-LPS, alterations in DNA methylation were the likely source of changes in

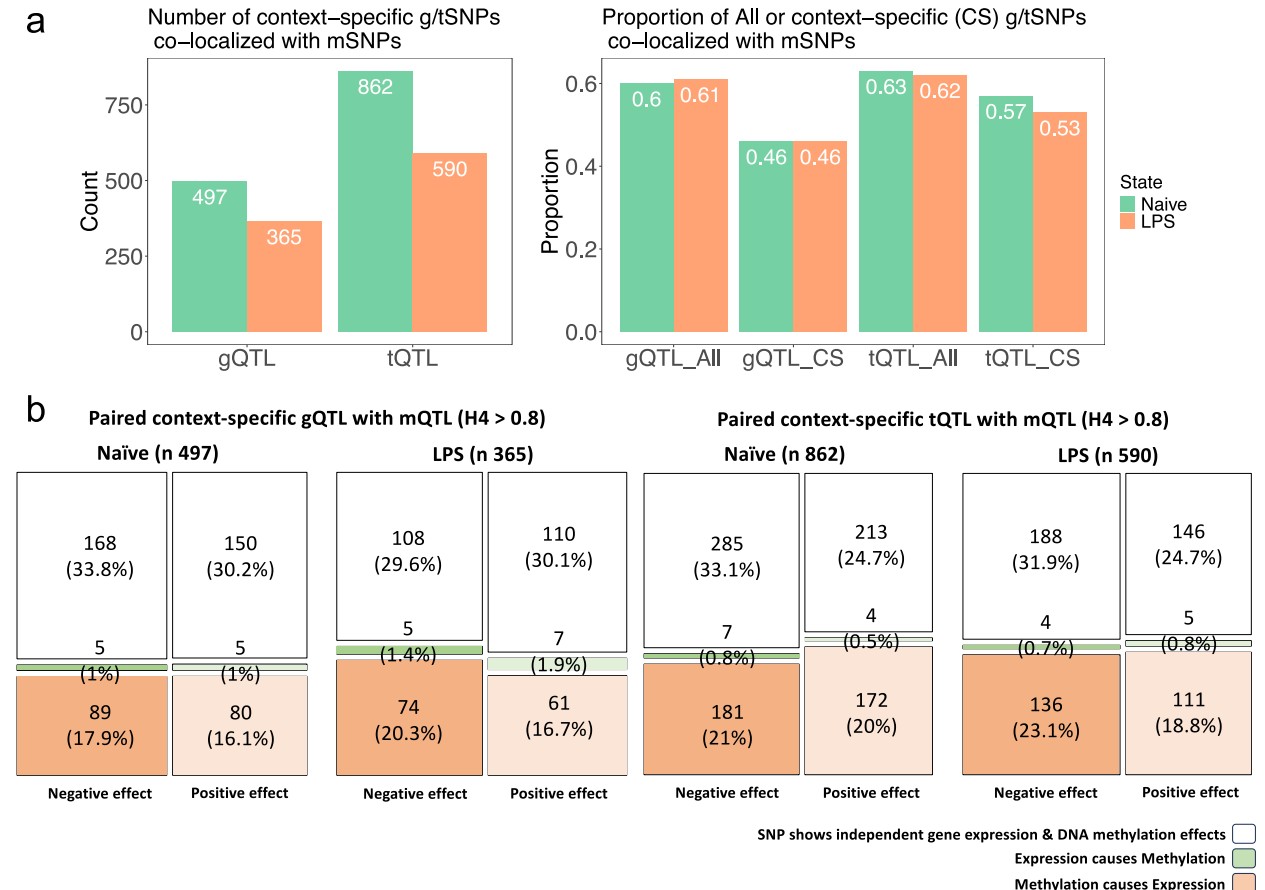

**Fig. 4 | Interaction between DNA methylation and expression. a** The number and proportion of g/tQTLs that are associated with mSNPs in naïve and LPS-stimulated monocytes (posterior probabilities of PPH4 > 0.8, FDR < 1 × 10⁻⁶). **b** A Steiger directionality test was conducted to determine the direction of causality between DNA methylation and gene/transcript expression. Our results indicated that the causal direction was most likely opposite, suggesting that increases in

expression were linked to reduced methylation. This finding is in line with the hypothesis that hypermethylation decreases the expression level in most cases. The area of boxes in the mosaic plot is proportional in size to the number of observed counts. Abbreviations: "n" represents and lists the specific number of g/tQTL-mQTL pairs, *CS* 'Context Specific'.

transcript expression. Conversely, in around 1% of the colocalised pairs of tQTLs and mQTLs (1.3% naïve and 1.5% post-LPS), alterations in transcript expression appeared to be the primary causal factor influencing DNA methylation (Fig. 4b). Our research showed that changes in DNA methylation and gene/transcript expression could be correlated in either a positive or negative way (Fig. 4b). In tQTL, an increase in methylation corresponded to a decrease in expression in 29% of naïve and 30% of post-LPS (negative correlation). In 25% of naïve cells and 26.8% of post-LPS, increased methylation was associated with an increase in expression (positive correlation). Figure 4b shows that context-specific gQTL and mQTL share similar patterns and proportions.

We present the LPS-specific, independent, and causal tSNPs that associate to transcript and gene expression, as well as the methylation level of CpGs (PPH4 > 0.8, FDR < 10⁻⁵, Fig. 5a). One notable example of such a shared g/tQTL and mQTL is the gQTL for *CD55* (encoding a crucial cell surface regulator of complement activation)[44] at rs2914937 (chr1:207315423 G > A) (Fig. 5b–d). rs2914937 is also a context-specific tQTL for *CD55* transcripts (*ENST00000367063* and *ENST00000314754*) (Fig. 5e–g), and an mQTL for cg22687766 within the upstream promoter across both resting and LPS-treated states. Colocalization analysis of the mQTL and gQTL associations was consistent with a shared causal locus at rs2914937 for *CD55* expression and cg22687766 methylation post-LPS (PPH4 = 1). These findings point to a single regulatory locus linking gene expression and epigenetic modifications in this region. Other

examples include rs6591507, a promoter genetic variant at chromosome 11p12, where the minor allele is associated with lower methylation level in the cg07745373 site within the same regulatory region and increased expression of *DTX4*, encoding an E3 ubiquitin ligase, induced innate immune stimuli (*P* = 1 × 10⁻¹⁸, Supplementary Fig. 7a–d).

Next, we systematically compared the stimulus-specific mQTL-associated lead SNPs against the GWAS summary statistics of traits. We observed moderate associations with autoimmune disorders and different cancer subtypes, indicating that DNA methylation may mediate some of the genetic risk of inflammatory disease processes, including cancer (Supplementary Fig. 8a). We observed that mSNPs specific to contexts are significantly enriched with "open transcript chromatin states" and "enhancer regions", in keeping with their roles in modulating the activity of regulatory elements (Supplementary Fig. 8b, c and Supplementary Data 4).

## The genetic determinants of gene regulatory network relationships

Genes are typically expressed in coordinated networks (Gene Regulatory Networks - GRN), and *cis*-acting polymorphisms may disrupt the relationship between *cis*-regulated genes and their associated GRN. By testing for divergent allele-specific correlation between *cis* genes and other GRN members, we sought to identify such gQTL, which we refer to as co-expression-QTL (coExQTL)[7,15]. Differential co-expression

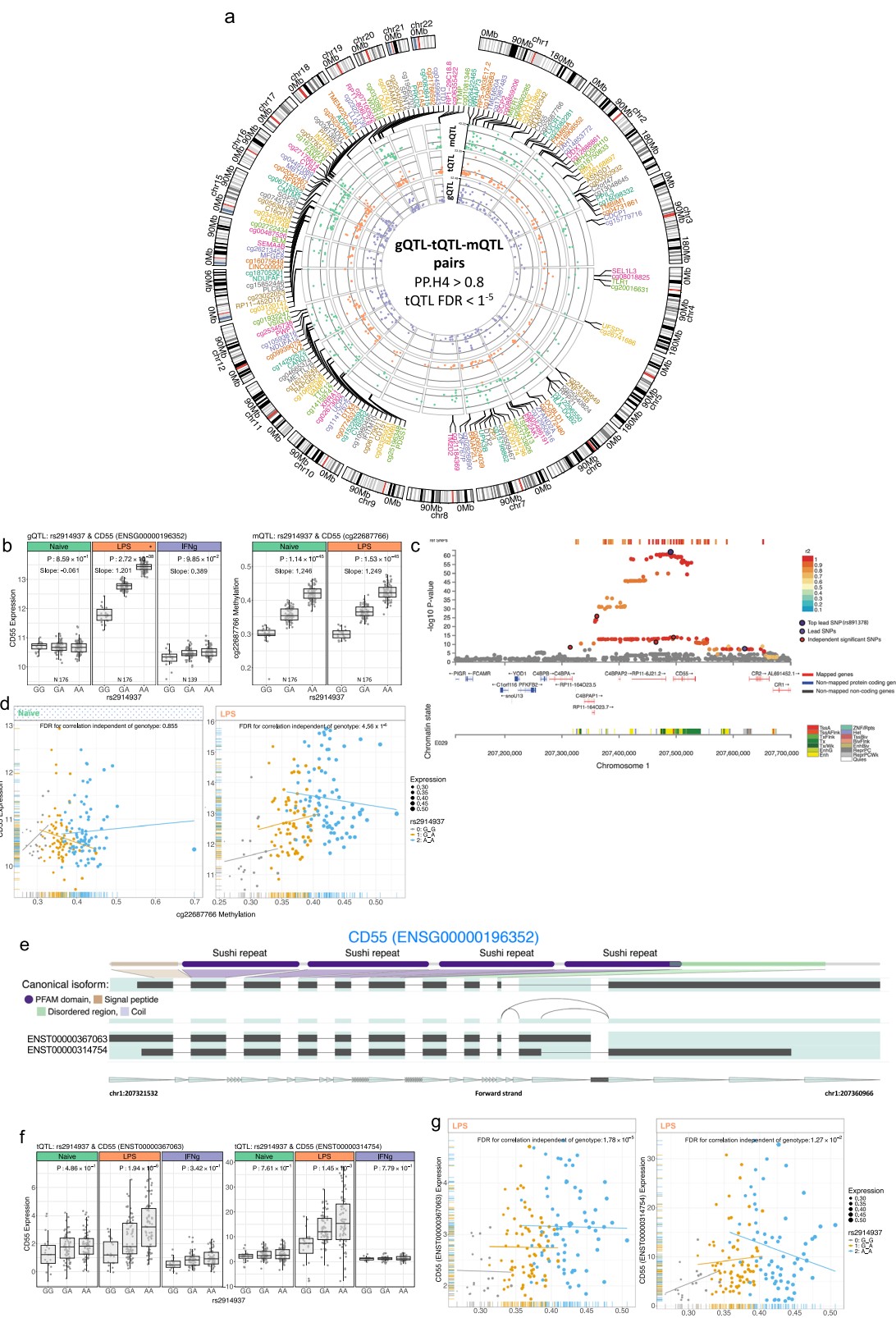

analysis of genes regulated by peak-gQTLs was performed with all other genes on an allelic basis to assess the significance of differences in correlation coefficients between correlated genes across genotype groups, with correction for multiple testing. To determine the biological significance of genes that are co-regulated, we conducted pathway analysis with HC2Allv2024 as our reference gene set[45,46]. To validate candidate coExQTL, we attempted replication using earlier independent microarray-based analysis of monocytes performed in the same conditions but in a different population[3]. Our downstream analysis was conducted on coExQTLs that demonstrated a similar gene pair, SNP, and direction of allele-specific co-expression relationship across both RNA-seq and microarray[3] datasets.

Across naïve, LPS and IFN-γ conditions, we identified 76, 41, and 75 candidate coExQTL, respectively, involving 4744, 4920 and 7026

**Fig. 5 | Interaction between DNA methylation and expression. a** Circos plot of LPS-specific g/tQTL-mQTL pairs that share one causal variant affecting both gene/transcript expression and methylation level. Gene/transcript and methylation sites are present on each track in g/tQTL-mQTL pairs of monocytes that are stimulated by LPS. Different colours are utilised to enhance readability. Ribbons link genes, transcripts, and methylation sites that have a causal variant together. The chromosome and genomic coordinates should be indicated on the outer track. **b** rs2914937 influences the methylation level (mQTL) and gene expression (gQTL) in LPS-stimulated monocytes. **c** Read coverage plot for *CD55* gQTL in LPS-stimulated monocytes. The lead SNP (rs891378) is in LD with rs2914937 (R² 0.99). **d** Expression of *CD55* correlates with the cg22687766 methylation site. The *CD55* expression and cg22687766 methylation, coloured by SNP rs2914937 genotype. Each data point on the horizontal and vertical axis indicates values for a single sample. Regression lines are shown for categories of genotypes. The adjusted *p*-value for the beta coefficient correlation between methylation and expression independent of the genotype is

$4.56 \times 1^{-6}$. **e** Highlighted isoforms of *CD55,* including *ENST00000367063* and *ENST00000314754,* show a significant switch in isoform usage across the LPS-stimulated condition. The display of splice junctions is limited to those that are not present in all transcripts. **f** rs2914937 influences *CD55* transcripts *ENST00000367063* and *ENST00000314754* most significantly post LPS. **g** Expression of *CD55* transcripts (*ENST00000367063* and *ENST00000314754*) correlates with the cg22687766 methylation site. The transcript expression and cg22687766 methylation level are coloured by the genotype SNP rs2914937. Abbreviations: "N" represents and lists the specific number of donors. The *x*-axis of the boxplot represents the different alleles of the SNP that are related to the expression. Each box indicates a particular SNP allele, and the *y*-axis displays the expression levels of individuals with that allele. The normalised read count and Fragments Per Kilobase Million (FPKM) metrics display the expression levels of genes and transcripts, respectively.

allele-specific co-expression relationships ($P_{interaction} < 10^{-6}$). To maintain consistency across platforms, we intersected genes in expression profiles from the RNA-seq and microarray datasets. The replication analysis was restricted to coExQTLs that involved genes present in this intersection (11371 genes). Using this shared gene set, we found 62, 34, and 60 candidate coExQTL across naïve, LPS, and IFN-γ conditions, involving 3761, 4029, and 5632 allele-specific co-expression relationships (Supplementary Data 7). Of these, we could replicate 69 (2%), 1054 (26.1%) and 1818 (32.2%) allele-specific co-expression relationships using array data (Supplementary Data 7). The majority of replicated allele-specific co-expression relationships were context-specific (UT: 15/69 (21.7%), LPS: 1053/1054 (99.9%), and IFN-γ (99.5%); $P_{interaction} < 0.05$). The lower rate of replication of co-expression relationships in naïve monocytes possibly reflects monocytes not being incubated in the earlier study[3].

Examples of coExQTL include rs7305461, a *cis* gQTL to *RPS26*, encoding ribosomal protein S26, a component of the 40S ribosome. *RPS26* is ubiquitously expressed and is mutated in Diamond-Blackfan anaemia, as well as having a large effect-size eQTL and being linked to numerous traits, including atopic disease[47]. Strikingly, in naïve monocytes, 171 allele-specific divergent gene correlations with *RPS26* were identified ($P_{interaction} < 0.05$), the most significant affecting the relationship between expression of *RPS26* and *KPNA2* (Fig. 6a), encoding a gene implicated in nuclear trafficking, indicative of a link between nuclear trafficking and ribosomal biogenesis (Fig. 6b). Pathway-based differential co-expression test of this coExQTL indicated rs7305461 disrupted the relationship between *RPS26* expression and GRNs involving protein secretion, cellular response to starvation, and response to viral infection (Fig. 6c and Supplementary Data 7). Whilst a physical interaction between the proteins encoded by *KPNA2* and *RPS26* has not been described, the encoded protein KPNA2 has been shown to bind RPS10, another component of the 40S ribosome, and similarly mutated in Diamond-Blackfan[48,49].

Subsequently, we explored the link between DNA methylation and coExQTL. In naïve monocytes, 12 gQTLs with replicated coExQTLs were shown to have allele-specific correlation with 1472 DNA methylation sites ($P_{interaction} < 0.01$). An example of this type of relationship is observed between the *RPS26* coExQTL, rs7305461, and 1459 methylation sites, including at cg10762038 ($P_{interaction} < 1.11 \times 10^{-5}$), which is correlated with *IFT20* from the coExQTL network that is linked to *RPS26* expression (Fig. 6d, c and Supplementary Data 8). cg10762038 and *IFT20* are located on chromosome 17 with a distance of 1.5 megabase pairs (correlation P 0.001), implying long-range enhancer activity of the cg10762038 locus[50,51]. The genetic variation that supports this coExQTL also likely has pleiotropic epigenetic effects, potentially indicative of divergent chromatin accessibility.

An example of an LPS-specific coExQTL is rs3110426, which disrupts a consensus *ZNF6* (Zinc Finger Protein 6)[52] binding site *cis* to *OXR1*, encoding Oxidation Resistance 1, involved in protection against

oxidative stress[53,54] (Fig. 6e). Here, 1049 allele-specific co-expression relationships were identified, of which 395 (37.6%) replicated in the same direction ($P_{interaction} < 10^{-6}$). The highest divergent correlation was noted with *PTGS2* which encodes COX-2, a protein of key importance in acute inflammatory responses and a leading pharmacological target (Fig. 6f). Notably, pathway-based differential co-expression test indicated a significant disruption of systemic autoimmune disease TNF-α signalling, bacterial infection pathways, and antigen processing cross presentation (FDR < 0.05) by allele (Fig. 6g and Supplemental Data 7).

In LPS-stimulated monocytes, 21 gQTLs had replicated coExQTLs that demonstrated allele-specific correlation with 857 DNA methylation sites ($P_{interaction} < 0.01$). An example of this type of relationship is observed between the *OXR1* coExQTL, rs3110426, and 76 methylation sites, including at cg24757533 ($P_{interaction} < 5.7 \times 10^{-6}$), which is correlated with 4 genes from the coExQTL network that are linked to *OXR1* expression (Fig. 6h, g and Supplementary Data 8). cg24757533 and correlated genes from the *OXR1* coExQTL network are located on chromosome 17 with a distance of 2 megabase pairs (correlation P values ranging from $10^{-3}$ to $10^{-23}$), again implying long-range regulatory activity[51,55].

Post IFN-γ the most robust coExQTL was rs2910789, forming a highly significant gQTL to *ERAP2*, encoding an endoplasmic reticulum aminopeptidase of key importance to antigen presentation, across all conditions (Fig. 6i). We observed 1576 genes to correlate with *ERAP2* expression in a genotype specific manner, of which 764 (48.4%) replicated in the same genotypically divergent direction, the most significant being *EPM2AIP1* (Fig. 6j). rs2910792 is r² 0.94 and D' 0.99 to rs2910686 which is associated with Crohn's disease and ankylosing spondylitis[56]. Pathway-based differential co-expression analysis demonstrated significant disruption of complement, and antigen processing/ ubiquitination-proteasome degradation (Fig. 6k, Supplemental Data 7). Interestingly, pathway analysis of genes involved with this coExQTL highlighted disengagement of the minor allele from GRN regulatory modules enriched in genes involved in mitochondrial activity (P $1.03 \times 10^{-10}$) and *MYC* targets (P $1.86 \times 10^{-05}$) (Fig. 6k and Supplementary Data 7). To further understand this, we explored genotype-specific correlation between *ERAP2* expression and mitochondrial count by performing quantitative PCR on mitochondrial and nuclear DNA extracted from monocytes in the earlier microarray-based analysis. Whilst mitochondrial count was not associated with age or sex of the donor, genotype specific divergent correlations between *ERAP2* expression and mitochondrial count were found (Fig. 6l). In individuals homozygous for the minor allele there was a positive correlation between *ERAP2* expression and mitochondrial count, whereas heterozygotes and homozygotes for the major allele showed a negative relationship between increased *ERAP2* expression and mitochondrial count. In the same analysis, we observed that mitochondrial count was highly associated with *MYC*

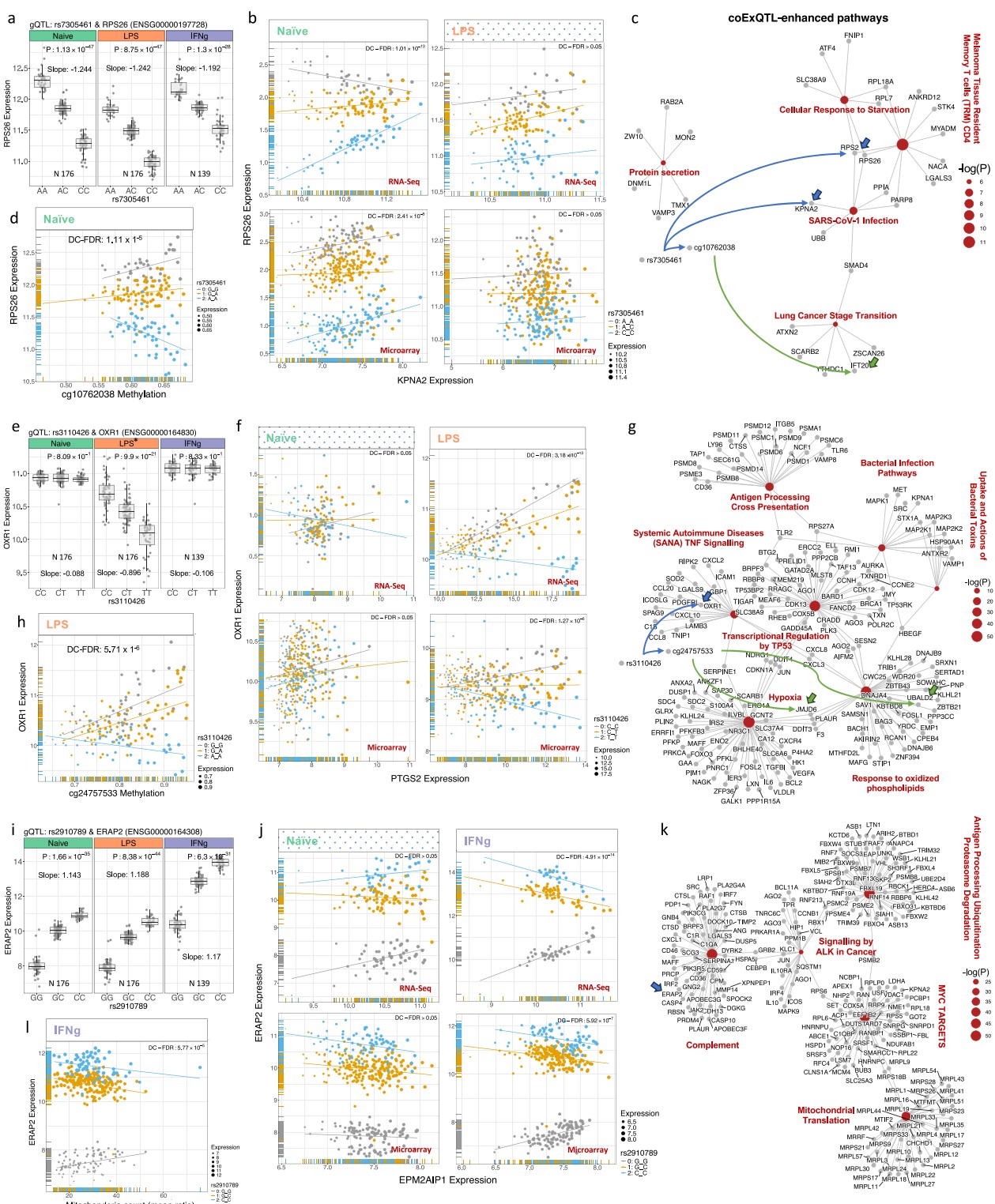

expression (P $1.32 \times 10^{-14}$, Supplementary Fig. 9 and Supplementary Data 7). Thus, this disease-associated locus is associated with apparent disruption of *ERAP2* expression from a MYC-regulated co-expression module that also regulates mitochondrial synthesis. Whilst ERAP2 plays a fundamental role peptide trimming for HLA loading, enabling antigen presentation[57], secondary associations with mitochondrial dysfunction have been described[58]. Moreover, a relationship between antigen presentation and mitochondrial demand has been demonstrated[59], as have links with the production

of reactive oxygen species[60] and pathogen engulfment[61]. Our data are in keeping with a genotype-specific manner association between *ERAP2* and *MYC* expression, which is associated with mitochondrial biogenesis[62].

## The genetic determinant of transcripts regulatory network relationships

Subsequently, we explored whether tQTL might similarly be associated with the uncoupling of GRNs to form coExQTLs. Across naïve, LPS and

**Fig. 6 | Condition-specific coExQTLs in stimulated monocytes. a** Local association plot for gQTL between the rs7305461 variant and expression of the *RPS26* gene across naïve and stimulated monocytes. **b** Identification of naïve-specific coExQTLs at rs7305461 (affecting the co-expression between *RPS26* and *KPNA2*, $P_{interaction} = 1.01 \times 10^{-13}$). The expression of *RPS26* and *KPNA2*, coloured by SNP rs7305461 genotype. **c** Pathway enrichment analysis of the coExQTL for *RPS26* and the rs7305461 variant. Pathway-based differential co-expression test was carried out for a group of genes that had allele-specific co-expression relationships for coExQTL at rs7305461 and were enriched with curated gene sets from online pathway databases. The genetic variant of rs7305461 is linked to cg10762038 methylation and *RPS26* expression. cg10762038 is correlated with *IFT20* (correlation P $3.51 \times 1^{-3}$) gene from the coExQTL network that are linked to *RPS26* expression. **d** The co-expression between genes can be explained by how rs7305461 affects both methylation and expression. A methylation site (cg07438246) that was linked to *RPS26* expression under the influence of a genetic variant has been demonstrated. **e** Post LPS, coExQTLs are present at rs3110426 and *OXR1* expression, with rs3110426 possibly affecting the binding site of ZNF628 transcription activator. **f** Context-specific co-expression QTL-rs3110426 between *OXR1* and *PTGS2*, $P_{interaction} = 6.98 \times 10^{-15}$, allelic expression coloured by SNP rs3110426 genotype. **g** Pathway enrichment analysis of the co-expression QTL for the rs3110426 variant. The genetic variant of rs3110426 is linked to cg24757533 methylation and *OXR1* expression. cg24757533 ($P_{interaction} < 5.7 \times 10^{-6}$) is correlated with *JMJD6* (correlation P $1^{-23}$) and *UBALD2* (correlation P $1^{-17}$) genes from the coExQTL network that are linked to *OXR1* expression. **h** The effect of rs7305461 on both methylation and expression can explain the co-expression between genes. A methylation site (cg24757533) that was linked to *OXR1* expression was found to be influenced by rs3110426. **i** coExQTL formed post IFN-γ include between the rs2910789 variant *ERAP2*. **j** Top condition-specific coExQTL-rs2910789 demonstrating co-expression between *ERAP2* and *EPM2AIP1* ($P_{interaction} = 7.09 \times 10^{-16}$), allelic expression coloured by SNP rs2910789 genotype. **k** Pathway analysis of *ERAP2* co-expressed genes demonstrates highly significant associations with the mitochondria pathway and *MYC* expression. **l** *ERAP2* expression correlates in an allele-specific manner with mitochondrial count, with rs2910789 influencing the relationship between mitochondrial count and *ERAP2* gene expression in IFN-γ-stimulated monocytes.

IFN-γ conditions, we identified 721, 900, and 505 transcript modules of coExQTL, respectively, involving 2969, 9621, and 2780 allele-specific co-expression relationships with $P_{interaction} < 10^{-5}$ (Supplementary Data 9). We found that transcript-level coExQTLs were frequently context-specific (UT: 521 out of 721 total, 72.2%; LPS: 653 out of 900 total, 72.5%; IFN-γ: 333 out of 505 total, 65.9%), indicating tQTL can be associated with disruption of GRNs. Context-specific transcript-level coExQTL analysis indicated that tSNPs specifically influencing the co-expression of a single transcript were the predominant mode of regulation in 98.4% of cases (1480/1507). In the remaining instances, where different tSNPs regulated the co-expression of multiple transcripts from the same gene, we found that 34.7% (8/23) of these exhibited overlaps in their upstream coExQTL genes, suggesting some shared regulatory mechanisms.

A key example was observed at a promoter tQTL (*ENSR00000753157*), rs10512696, which is associated with *ENST00000503513*, a transcript variant of the *DAB2* gene characterised by different first exon usage, intron retention, and alternative 3' splicing, leading to divergent protein structure across all contexts. In the naïve state, this variant also forms a coExQTL with 155 allele-specific co-expression relationships (Fig. 7a–d). The strongest co-expression relationship for *ENST00000503513* was with *PPP1CA* (*ENSG00000172531*) (P interaction = $6.18 \times 10^{-7}$), encoding a PP1A, a phosphatase with a central role in signalling cascades[63,64]. Correspondingly, pathway-based differential co-expression test of this coExQTL indicated that rs10512696 disrupts the relationship between *DAB2* expression and GRNs involving TGFB1 targets, mitochondria, and genes involved in the cellular response to chemical stress (Fig. 7e and Supplementary Data 9).

Post-LPS rs61869825 disrupted coordinated expression of 168 canonical transcripts with *ENST00000337003* ($P_{interaction} < 0.001$), a transcript variant of the *USMG5* gene encoding a component of the mitochondrial ATP synthase complex (Fig. 7f, g). Notably, we observed a strong association between rs61869825 *ENST00000337003* and expression of *LINC00339*, a long non-coding RNA associated with cancer invasion[65] (Fig. 7g). The associated coExQTLs were enriched in key biological pathways, including responses to LPS, hypoxia, and IL4 (Fig. 7i, Supplementary Data 9).

Finally, we identified 505 coExQTL comprising 2780 allele-specific co-expression relationships post-IFN-γ. We observed 491 independent tSNPs related to these modules, indicating a significant genetic impact on the coordinated expression of these gene isoforms (Supplementary Data 9). Notably, we found the previous tQTL activity at *OAS1* (*ENST00000445409*) was associated with allele-specific co-expression forming coExQTL ($P_{interaction} < 10^{-5}$) (Fig. 7j–m). Most significant being context-specific *ENST00000445409* activity between *OAS1* and *IGFBP7* post-IFN-γ (Fig. 7j–l) with pathway enrichment highlighting antiviral mechanisms (FDR < 0.05) (Fig. 7m and Supplementary Data 9).

## Discussion

Our observations across three divergent conditions of immune stimulation provide further insights into the relationship between primary monocyte activation state and genetic regulation of gene expression and splicing. The integration of summary data from GWAS and both tQTL and gQTL analyses highlights the relevance of our findings to conditions including autoimmune diseases[66,67], cardiovascular risk factors[68], neurodegeneration[69] and severe COVID-19[10]. Mounting evidence suggests that in many of these conditions, often thought to be primarily lymphoid in pathogenesis, monocytes play a key role secondary to the release of pro-inflammatory mediators and antigen presentation[70,71]. The study will be a resource for those interested in monocyte splicing in immune activation.

Our focus has been to address the gap in our understanding of the effect of environmental factors relevant to infection and inflammation on genetic determinants of expression at the transcript level[69,72]. Our work adds to previous studies[3,68] by introducing transcripts that had monocyte expression affected by multiple *cis*-acting tSNPs. Independent SNP effects were responsible for about 3.73% (390/10452) of the detected tQTLs. It should be mentioned that 3.84% (15/390) of the transcripts were affected by multiple context-specific tSNPs, which were previously known to be located at disease-associated loci (FDR < $10^{-5}$). It is possible that the disease susceptibility could be influenced by multiple genetic effects in a specific cell or environmental context. The relevance of such tQTLs in disease is demonstrated by our observation of rs57484342 at the *OAS1* COVID-19 severity risk locus associated with divergent *OAS1* splicing post-IFN-γ exposure, a condition akin to early-onset viral infection[10].

By integrating observations with DNA methylation data from the same individuals, we provide further insights into the relationship between genetic variation, DNA methylation and expression. We frequently observe directional causal effects between regulatory variants inferred to regulate expression secondary to the effects on methylation and vice versa. Again, these mQTL-gQTL pairs are observed to colocalise with GWAS disease risk loci, and we anticipate that these data will be further used by the wider research community to map causal variation.

Finally, how gQTL and tQTL intersect with regulatory networks is less well characterised, but it is key to understanding the complex impact of regulatory variation on phenotype. We reasoned that *cis*-acting variants may disengage genes where expression is coordinated with other genes in the same biological pathway[3,16], impacting the composition of GRNs. In such cases, we expect evidence of the genetic variant's regulatory influence on the gene-gene interaction in the form of genotype-specific differential gene correlation[16,46]. This work adds to previous studies exploring the effect of gene co-expression in an allele-specific manner[3,16], the effect of *cis* genetic variants on

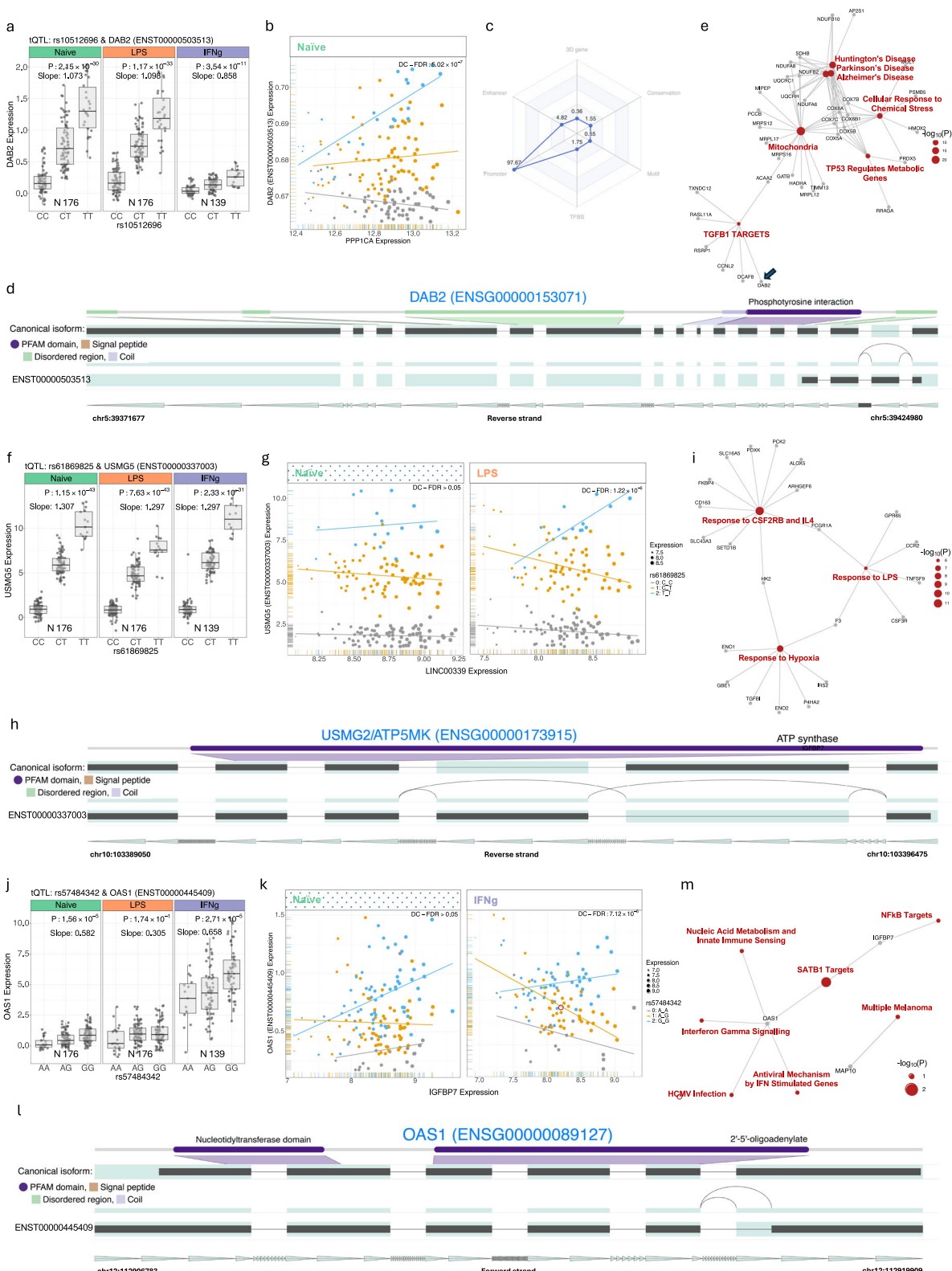

modulating gene expression response[3], and cell type-specific *cis*-g/tQTL and co-expression QTL[15,73]. By defining coExQTL, which we show to be intricately associated with divergent pathways of gene expression, we extend on previous studies constructing condition-specific correlation networks[7,15,46,74–79]. Notably, many potential interactions may not be apparent at the gene level, since the impact of one allele may vary across different transcripts; however, differential transcript correlation analysis in an allele-specific manner has not been

systematically applied. By applying this approach to tQTL we identify many examples of similar changes in GRN associations at the transcript level, further illustrating the complexity of regulatory variant activity across contexts.

We note several important coExQTL, most strikingly at *ERAP2*, where rs2910792, (r² = 0.94 with rs2910686; CD risk allele) uncouples *ERAP2* expression from a *MYC*-regulated pathway incorporating multiple components of mitochondrial synthesis. Our observations are

**Fig. 7 | coExQTL analysis reveals significant associations between genetic variants and the coordinated expression of transcript modules. a** Local association plots for tQTLs between the rs10512696 variant and expression of the *DAB2* isoform (*ENST00000503513*) across naïve and stimulated monocytes. **b** Identification of naïve-specific coExQTLs at rs10512696, which affects the co-expression between *ENST00000503513* and *PPP1CA*. The expression is coloured by the genotype SNP rs10512696. Each data point on the horizontal and vertical axis indicates values for a single sample. **c** Functional significance score was applied to evaluate the functional significance of rs10512696 in 6 categories. **d** The figure shows the isoforms landscape of *DAB2*. **e** Pathway enrichment analysis of the coExQTL module of *DAB2* in naïve monocytes. Pathway-based differential co-expression test of the coExQTL module indicated that the average correlation change between genes in the genotype classes of rs10512696 was statistically significant (FDR < 0.05). **f** Local association plots for tQTLs between the rs61869825 variant and the expression of the *USMG5* isoform (*ENST00000337003*). **g** The LPS-specific coExQTL at rs61869825 affects the co-expression between *ENST00000337003* and LINC00339 (*ENSG00000218510*). **h** *USMG5* (canonical and *ENST00000337003*) isoforms are

highlighted in the isoform landscape plot. **i** The pathway analysis of context-specific coExQTL modules in LPS24 monocytes shows that they are significantly associated with multiple pathways, including the response to aggregations of unfolded proteins. **f** Local association plots for tQTLs between the rs61869825 variant and the expression of the *USMG5* isoform (*ENST00000337003*). **g** LPS-specific coExQTL at rs61869825, which affects the co-expression between *ENST00000337003* and *LINC00339* (ENSG00000218510). **h** The isoform landscape plot highlights isoforms of *USMG5* (canonical and ENST00000337003). **i** Pathway analysis of coExQTL module of *USMG5* in LPS24 monocytes. **j** Local association plots for tQTLs between the rs57484342 variant and the expression of the *OAS1* (*ENST00000445409*). **k** Naïve-specific coExQTL at rs57484342, which affects the co-expression between *ENST00000445409* and *IGFBP7*. **l** The isoform landscape plot highlights isoforms of *OAS1* (canonical and *ENST00000445409*). **m** Pathway analysis of the coExQTL module of *OAS1* in IFN-γ stimulated monocytes. Abbreviations: "*N*" represents and lists the specific number of donors, *TFBS* transcription factor binding site.

---

particularly intriguing because of the different disease associations at this locus, particularly in relation to inflammatory and infectious diseases[80]. Furthermore, we discovered that coExQTL genes are more likely to be identified through transcript-level coExQTLs. A key example being coExQTL formation at the *OAS1* locus according to rs57484342 carriage.

By integrating gene-level coExQTL analysis with DNA methylation profiling in the naïve and LPS states, we further illustrate the utility of multiple layers of genomic information to uncover complex regulatory mechanisms. This was exemplified by the identification of an mQTL at the promoter for *RPS26* linking an mQTL to the observed coExQTL. While our results in this smaller dataset are limited, we think that incorporating this approach into larger datasets could uncover genotype-dependent GRN relationships.

While our findings provide valuable insights into the impact of immune conditions on regulatory genetics in monocytes, it is important to acknowledge limitations. The complex cytokine milieu and cell-cell interactions that characterise the immune system in both health and disease cannot be faithfully replicated in vitro and require multiple approaches to fully dissect. Whilst there are strengths with single treatment models, it is increasingly important to utilise patient-derived data in the relevant disease state. Such work, whilst in its nascent stage, is providing insights into the role regulatory variation plays in sepsis immunity and response to immunotherapies[81,82]. Whilst bulk RNA-sequencing provides the depth of sequencing required to provide a comprehensive overview of monocyte expression and splicing across conditions, this work is at the expense of determining the resolution of genetic effects that vary by cellular subset[7]. Although short reads and sparsity of gene coverage are limitations for such analysis at the single-cell level, rapid advances in this domain and single-cell methylation analysis are envisaged to permit less-sparse datasets and characterisation of splicing to complement these data. Despite these limitations, our study provides a comprehensive reference map of genetic influences on regulatory monocyte expression and splicing in conjunction with DNA methylation across divergent innate immune activity that will help further elucidate the contribution of monocytes to disease aetiology. All data generated by this study is made available for further exploration and analysis, and we provide a web-based database to enable researchers to conveniently access specific observations.

## Methods

### Ethics

Blood for monocyte isolation was taken from donor participants who were recruited via the Oxford Biobank (www.oxfordbiobank.org.uk; ethical approval reference 06/Q1605/55), having provided written, informed consent.

## Data generation

**Sample preparation, RNA isolation and sequencing.** Peripheral blood mononuclear cells (PBMCs) were isolated from 192 healthy individuals of European ancestry. Blood cells were separated from freshly drawn blood using Ficoll gradient purification. Monocytes subsequently positively were selected using magnetic CD14[+] isolation kits (Miltenyi) according to manufacturer's protocols, and cell purity was found to be a median > 99%[12]. Monocytes were cultured at 500,000 cells per mL in 400 μL RPMI supplemented with L-Glutamine, Penicillin/Streptomycin and 20% FCS in BD Falcon 5 mL polypropylene culture tubes. Post purification samples were rested overnight at 37 °C, 5% CO2 prior to being further incubated for 24h alone (UT) or in the presence of 20ng/ml LPS (Ultrapure LPS, Invivogen) or 20ng/ml IFN-γ (R&D Systems). Poly-A RNA was paired-end 100 bp sequenced in the Oxford Genome Centre using Illumina Hiseq-4000 machines. 506 high-quality transcriptomes (mean ~50 million reads) were mapped (188 Untreated, 188 LPS, 144 IFN-γ).

The methylation profile of naïve and LPS-stimulated primary monocytes from 176 individuals were assessed using the Illumina 450 K array, which quantified methylation levels at 300,885 CpG dinucleotides. We excluded 96,427 loci that were analysed using probes that contained SNP(s) at/near the targeted CpG site (≤ 50 base pair), as these may not be enough to measure DNA methylation levels[83].

In IFN-stimulated monocytes, the ratio of mitochondrial DNA (mtDNA) to nuclear DNA (nDNA) was used to estimate the relative abundance of mitochondria. The amount of mtDNA and nDNA in the samples was quantified by performing qPCR after extracting DNA from monocyte samples. To calculate the mean ratio, the amount of mtDNA was divided by the amount of nDNA.

**Sample size calculation of bulk tissue g/tQTL analysis.** The power-EQTLSLR R function was utilised to calculate the power for g/tQTL analysis[84]. By supplying values for sample size, minimum detectable slope, standard deviation of the outcome (y) in simple linear regression (*sigma.y*), and minimum allowable MAF parameters, this function can be utilised to calculate power. Power is used to determine the likelihood of accurately detecting a real association between a genetic variant and gene/transcript expression. If a true association is present, a higher power means a better chance of detecting it.

The *sigma.y* can be calculated as $sigma.y = slope/\sqrt{2 \times MAF \times (1 - MAF)}$. The slope of the simple linear regression parameter from our tQTL/gQTL was adjusted to 0.7, and the MAF was set to 0.04 from ref. [3]. The estimated testing power for a sample size of 138, with $a = 0.2$ and family-wise error rate (FWER) = 0.01, was 1.

**Genotyping and genotype imputation.** Genotyping was performed with Illumina HumanOmniExpress with a coverage of 733,202 separate

markers. Analysis of identity-by-descent was performed using PLINK[85], which demonstrated that there is no shared genetic material between the individuals (PI_HAT ranged from 0-0.047, median 0). Genotypes were pre-phased with SHAPEIT2, and missing genotypes were imputed with PBWT[86], vcftools (v0.1.12b) was applied to genetic variation data in the form of variant call format (VCF) files to filter out indels and SNPs with minor allele frequency less than 0.04[87]. We used the CrossMap tool for the conversion of coordinates between genome assemblies[88].

### Data processing

**Quantification and gene expression analysis.** Sequencing reads were aligned to CRGh38/hg38 using HISAT2 for each sample individually, and default parameters. High mapping quality reads were selected based on the MAPQ score using bamtools. Marking and removing duplicate reads were performed using Picard (v 1.105)[89]. Samtools was used to pass through the mapped reads and calculate statistics[90]. We detected sample contamination and swaps based on a comparison of the imputed SNP-array genotypes with genotypes called from RNA-seq using verifyBamID[91].

491 high-quality transcriptomes from 185 individuals (properly paired = 30,992,754,324 reads, median = 47,735,438) were selected and used for downstream analysis (176 Untreated, 176 LPS, 139 IFN-γ) (Table 1).

Gene read count information was generated using HTSeq[92] (v 2.0.5), and lowly expressed genes (those with < 50 reads) were filtered prior to applying conditional quantile normalisation, resulting in 15641 genes for analysis. Variance stabilising transformation was subsequently applied to the matrix using DESeq2 (v 1.18.1)[93] to normalise gene read counts, yielding approximately homoscedastic profiles[94,95]. Assembly of the alignments into full and partial length transcripts and transcript-level expression analysis of RNA-seq experiments were done using StringTie[96]. The expression values of a uniform set of 27540 transcripts with minimum input transcript FPKM (Fragments Per Kilobase of transcript per Million mapped reads) ≥ 0.5 for all individuals of at least one of the treatments were applied for downstream analyses[57]. We applied the IsoformSwitchAnalyzeR tool to analyse isoform usage of 16198 genes, including 84986 transcripts in naïve and treated monocytes with IFN-γ or LPS[97]. Isoform usage refers to the fraction value of the mean isoform expression given the mean expression of the corresponding gene in a setting with $k$ biological replicates[98].

**gQTL, tQTL and mQTL analyses.** We studied the association of the variant with alternative splicing using complementary steps including gene expression QTL (gQTL), transcript QTL (tQTL), and methylation QTL (mQTL).

The normalised total gene counting sequencing read or transcripts expression values (FPKM) were regressed against genetics variants. SNPs were included in the cis analysis if they were located within 1 Mb of the gene or 100 Kb of the isoform under consideration[32]. By reducing multi-test burden, smaller windows, such as 100 Kb centred on the transcripts start site, can improve power[20].

We decomposed the gene expression matrix to the loading and score matrices. The score matrix was applied as a covariate of the Linear Model to adjust for unexplained variation in gene expression (observed dependent variable) and reveal the actual effect of genetics (categorical independent variable). Zero to 50 principal components (PCs) of gene expression profiles were tested using a total of 1000 permutations[99], to determine the optimal number of PCs that capture the most significant variation in the data without overfitting. Technical noise, batch effects, or other confounding factors that can affect gene expression measurements can often be captured by dominant PCs. In the regression model for g/tQTL analysis, we used the dominant PCs as independent variables. In this regression, the residuals represent the gene expression levels that have been corrected for the effects of the dominant PCs.

**Table 1 | Summary statistics of properly paired reads in high-quality transcriptomes from 185 individuals per state**

|  | Naïve (*N* = 176) | LPS (*N* = 176) | IFN-γ (*N* = 139) |
| --- | --- | --- | --- |
| counting sequencing reads | 8,440,064,294 | 8,789,476,734 | 6,753,408,378 |
| Min. | 30,185,698 | 32,037,326 | 29,967,830 |
| 1st Qu. | 43,804,162 | 46,116,686 | 45,043,112 |
| Median | 47,762,042 | 50,015,776 | 47,540,954 |
| Mean | 47,954,911 | 49,940,209 | 48,585,672 |
| 3rd Qu. | 52,402,000 | 54,084,506 | 52,008,373 |
| Max. | 68,882,270 | 66,147,520 | 64,422,960 |

Inflection Points and Local Maxima were employed to select the most suitable number of PCs to be included as covariates in g/tQTL analysis (Supplementary Fig. 1a). To compare the number of PCs to the number of detected g/tQTL, we used a scree plot. Our next step was to discover a pronounced inflection point where the slope of the curve begins to decrease significantly. This could indicate that adding more PCs is not providing any significant additional information. The point where the curvature changes sign is called an inflection point on a curve. When it comes to PCA, it frequently indicates a significant change in the rate of variance explanation by additional PCs. Local maximum refers to a point on a curve where the function reaches its highest value within a given interval. In PCA, it represents a PC that explains a relatively large amount of variance.

The statistical power and balanced male/female distribution of our sample size were not sufficient to detect subtle effects. Due to this, g/tQTL analysis for sex-specific stimuli was not possible[93,100].

g/tQTL analysis was performed with the QTLtools using a linear regression[28]. We used QTLtools to calculate nominal g/tQTL summary statistics (https://github.com/francois-a/fastqtl). tQTL analysis requires multiple testing corrections at two levels: due to multiple transcripts per gene (molecular trait) and due to multiple molecular traits across the genome. We applied conditional analysis (implemented in QTLtools) for tQTL analysis of multiple molecular phenotypes (transcripts) belonging to higher-order biological entities (genes). We used the --permute 1000 and --grp-best options in QTLtools to calculate empirical *P*-values at the group level and estimate the standard error of the effect sizes, after a permutation pass on the data, was done[28]. The same procedure was applied to screen relationships of DNA methylation in response to the naïve and LPS-stimulated monocytes associated with local genetic variation within 1 Mb of the start site of the gene under consideration. We performed a permutation-based mQTL analysis on methylation data to adjust *P*-values for the number of methylation sites and genetic variants in cis given by the fitted beta distribution[56].

**Lead and independent SNP identification.** The SNP with the strongest association signal within a region is known as Lead SNP[8,101]. It is often the SNP that is most likely to be the causal variant, but this is not always true. Linkage disequilibrium (LD) analysis and clumping procedures were used to identify the lead SNPs within every associated locus. LD identifies groups of SNPs that are highly correlated by measuring the correlation between genetic variants. The clumping algorithm identify SNPs that have a high correlation with each other and groups them together[8]. This assists in prioritising the most likely causal variants within a region. The number of SNPs considered was reduced, while the most informative variants were retained by clumping together SNPs with high LD.

The input of the Linkage Disequilibrium (LD) analysis was a list of SNPs in gQTL (FDR < 0.01). First, the degree of similarity of all SNPs associated with an indicated phenotype was measured using Pearson's $\chi^2$-statistics and 1000 genome data in the European population

background[8]. Next, we clumped SNPs ($r^2 < 1^{-3}$) and for each region of high LD, kept the SNPs with the lowest $p$-value[8]. The application of independent of the underlying haplotype-block structure for identification of lead SNPs has been reviewed by ref. [101]. This approach made it possible for us to concentrate on the leading SNPs in every locus, which are likely to be the most functionally relevant variants that are driving the observed association. To investigate the shared genetic determinants of methylation and gene expression, we utilised lead gSNPs.

Independent-g/tQTL is a term used to describe a significant connection between an SNP and a gene/transcript expression level that persists after conditioning on other SNPs in the same genomic region[102]. We used conditional pass analysis in QTLtools[28] to detect independent-g/tQTLs. To detect independent signals (i.e., independent SNPs), each SNP in the region was sequentially conditioned, and the effect of the remaining SNPs on the trait of interest was evaluated. We conducted permutations per molecular phenotype and forward–backwards stepwise regression to assign all significant variants per cis-window and determine the most promising hit per independent signal[28]. After conditioning on the primary g/tQTLs, the independent g/tQTLs has an independent SNP that is closely linked to genes, transcripts, and methylation. This indicates that the SNP is most likely a genuine QTL and not just a proxy for other SNPs in proximity. Conditional pass analysis was performed to identify multiple proximal SNPs with independent effects on a molecular phenotype and specify context-specific gQTLs and tQTLs.

In order to balance sensitivity and specificity, we limited our analysis to independent-gQTLs with FDR < $10^{-6}$ and MAF > 0.039. The MAF was set to 0.04 from ref. [3]. Variants with an MAF below 0.04 may not be able to detect significant associations with gene expression due to insufficient statistical power. Low-frequency variants tend to have fewer carriers, which makes it more difficult to identify their effects. Variants with MAF below 0.04 are typically regarded as rarer in the population. Although they may still have significant biological effects, they may not be as likely to be involved in common phenotypic variations. Low-frequency variants can have a higher risk of genotyping errors because they are less frequently encountered by genotyping platforms. To minimise the impact of such errors on the analysis, a 0.04 threshold is utilised[3]. To guarantee the reliability of identified gene-expression associations, it is common to use FDR less than $10^{-5}$ in eQTL studies. Controlling false positives allows to concentrate on biologically meaningful and reproducible findings[103,104].

**Correction for multiple testing in g/tQTL analysis.** It's essential to deal with the problem of multiple testing correction when performing g/tQTL analysis with multiple transcripts per gene to prevent false discovery rate (FDR). The correction process involves two steps of multiple-testing, which involve separate FDR correction and combined FDR correction. Separate FDR correction computes association statistics for related transcripts and each variant independently for each gene. After that, the $p$-value at the gene level is calculated, which takes into account the number of transcripts and variants tested. The FDR at the transcript level can be controlled by this. QTLtools were used to identify primary eQTL for a methylation site (mQTL), gene (gQTL), or transcript (tQTL) of interest in order to perform separate FDR correction.

**Context-specific quantitative trait.** Context-specific quantitative trait loci are eQTL that are revealed after specific biological stimuli[3]. Of primary interest in context-specific eQTL analysis is identifying eQTLs for which the correlation within condition varies across conditions[3]. We denote such an eQTL in condition $k$ by $i^k$. When $K = n$ conditions are being considered, we say an eQTL maybe a context-specific eQTL for the condition $i$ if eQTL is just significant for condition $i$ and not for

$i^C$. Differential context-specific quantitative trait loci are context-specific eQTLs that effect of QTL alleles is revealed on more than one condition but with different directions [sgn($b^i$)≠sgn($b^{i^C}$) where $b$ represents the slope of a regression line].

Conditional analysis implemented in QTLtools[28] was performed to identify multiple proximal eQTLs with independent effects on a molecular phenotype and specify context-specific gQTLs and tQTLs. In a cis conditional pass, the independent signal indicates a significant association between a SNP and a quantitative trait that remains significant even after conditioning on other SNPs within the same genomic region. This suggests that the SNP is probably a genuine QTL and not just a proxy for other nearby SNPs. The process of detecting independent signals involves sequentially conditioning each SNP in the region and evaluating the effect of the remaining SNPs on the trait of interest[55]. To that end, we ran permutations per molecular phenotype and a forward–backwards stepwise regression to assign all significant variants per cis-window and determine the best candidate hits per independent signal. In addition, we applied approximate conditional analysis (moloc)[29] to specify context-specific gQTLs and tQTLs (PP > 0.5) across naïve and stimulated monocytes as described by[30]. Evidence for shared or independent effects of genetic variants can be compared using the Moloc tool. By studying the colocalization of g/tQTL, we were able to identify highly active g/tQTL in specific contexts.

**SNPs functional annotation and causal relationship analysis**
**Integration of GWAS trait-associated SNP and g/tQTL.** The 380 GWAS summary statistics from UK-Biobank[105] and MR Base GWAS databases[8] (European-ancestry individuals) containing significant associations were prepared as instruments of GWAS trait causal relationship analysis. We selected GWAS summary statistics in subcategories of autoimmune/inflammatory, cofactors/vitamins, haematological, immune system, cancer, immune cell subset frequency, metabolites, and nucleotide and used to infer causal relationships. The input of the analysis is a list of g/tSNP genotype data (MAF, reference and alternative alleles), effect size and p-value of g/tQTL analysis. To ensure that the query SNPs are independent, the European samples from the 1000 genomes project were used to estimate LD between SNPs and perform SNP clumping ($r^2 < 1^{-3}$ and MAF = 0.01). For a set of SNPs in the same LD block, only the SNP with the lowest p-value was retained. Next, we harmonised the reference and minor alleles of common SNPs between each GWAS summary statistics and query. If a particular query SNP was not present in the given GWAS summary statistics, then a causal LD proxy SNP was selected and searched for instead. A causal LD proxy SNP was defined as an SNP in LD with the query SNP and causal in both GWAS and g/tQTL/mQTL studies. We used colocalization tests of two genetic traits (coloc)[106] with default parameter for prior probabilities to estimate the Bayes factor as the posterior probability that an eSNP is causal in both GWAS and g/tQTL studies (Supplementary Fig. 11). Coloc examines the posterior probability of four hypotheses (H), including PP.H1.abf (SNP associated with gene expression only), PP.H2 (SNP associated with GWAS trait only), PP.H3 (SNP associated with neither trait), and PPH4 (SNP is associated with both gene expression and GWAS trait). The SNP's high value of PPH4 suggests that it could have a causal effect on both gene expression and GWAS trait, indicating a potential regulatory relationship[106]. To identify causal relationships in the region of interest, we utilised the PPH4 > 0.8[106].

The Mendelian randomisation (MR) test was applied to enrich the harmonised list of query SNPs and GWAS summary statistics (GWAS p-value threshold of $1^{-6}$)[107]. First, the effect sizes of the g/tQTL ($\beta_{SNP-eGene}$) were provided in stimulated or naïve monocyte samples. Next, the association between these same SNPs and traits were extracted from GWAS databases ($\beta_{SNP-Trait}$). These slopes of regression models were combined to yield estimates for each SNP of the effect of monocytes on a trait ($\beta_{exposure-outcome} = \frac{\beta_{SNP-outcome}}{\beta_{SNP-exposure}} = \frac{\beta_{SNP-trait}}{\beta_{SNP-eGene}}$).

**Table 2 | The 15 core chromatin state abbreviations are broken down**

|   | Abbreviation | Full name | Description |
|---|---|---|---|
| Transcription Start Site (TSS) Related States | | | |
| 1 | TssA | Active TSS | Indicates a region actively involved in transcription initiation. |
| 2 | TssAFlnk | Active TSS Flank | Describes a region flanking the active TSS, often associated with regulatory elements. |
| 3 | TxFlnk | Transcription | Refers to the process of gene transcription. |
| 4 | Tx | Transcription Flank | Indicates a region flanking a transcript, potentially containing regulatory elements. |
| 5 | TxWk | Weak Transcription | Suggests a region with low levels of transcription activity. |
| Enhancer-Related States | | | |
| 6 | EnhG | Gene Enhancer | Indicates an enhancer element associated with a specific gene. |
| 7 | Enh | Enhancer | A region of DNA that can boost the expression of a gene. |
| 8 | ZNF/Rpts | Zinc Finger/Repeats | A type of DNA binding motif often found in enhancers. |
| Heterochromatin and Bivalent States | | | |
| 9 | Het | Heterochromatin | Densely packed chromatin that is generally transcriptionally inactive. |
| 10 | TssBiv | Bivalent TSS | A TSS marked with both active and repressive histone modifications, often associated with developmental genes. |
| 11 | BivFlnk | Bivalent Flank | A region flanking a bivalent TSS, potentially containing regulatory elements. |
| 12 | EnhBiv | Bivalent Enhancer | An enhancer marked with both active and repressive histone modifications. |
| Repressive States | | | |
| 13 | ReprPC | Repressed Polycomb-repressed Complex | A region marked by repressive chromatin modifications mediated by Polycomb proteins. |
| 14 | ReprPCWk | Weak Repressed Polycomb-repressed Complex | A region with lower levels of Polycomb-mediated repression. |
| 15 | Quies | Quiescent chromatin | Refers to a state that is generally inactive in transcription. |

Finally, the $\beta_{exposure-outcome}$ estimates of query SNPs were averaged to produce a magnitude of the overall causal effect of monocytes on a trait[107].

To infer causal relationships between exposures (gene expression) and outcomes (traits), we employ the Mendelian randomisation test[108]. MR uses genetic variants as instrumental variables to estimate the causal effect of exposure on an outcome. The null hypothesis is calculated by MR based on the assumption that any correlation between the exposure and the outcome is caused by chance or confounding factors. MR utilises statistical methods like the inverse variance weighted method or the Wald ratio test to evaluate the evidence for a causal relationship between exposure and outcome. The p-value is determined by comparing the observed association between exposure and outcome and the null hypothesis. The strength of evidence against the null hypothesis and in favour of a causal relationship can be indicated by a smaller p-value.

We used the TwoSampleMR package (version 0.5.2) in R (version 4.3.1) to perform penalised weighted median MR analyses[108].

**Chromatin state and genomic features association SNP enrichment.** Fifteen-core chromatin states (Table 2) for primary monocytes peripheral blood (E029 chromatin states dataset provided by the Roadmap Epigenomics Consortium[14]) and 10 genomic features from the biomaRt (Ensembl) and UCSC[109] were used to annotate a list of SNPs in tQTLs, gQTLs, and mQTLs (FDR < 0.01, MAF > 0.039).

The biomaRt repository covers 409,304 regulatory features, including genomic coordination and associated sequence motifs of 139729 CTCF binding site, 74575 enhancers, 63152 open chromatin region, 25313 promoters, 87975 promoter flanking region, and 18560 transcription factor (TF) binding sites. UCSC repository provides annotations of 1,619,806 genomic features, including 64334 3'UTRs, 114271 5'UTRs, 659198 introns, 289809 exons, and 82890 promoters. We downloaded the annotation databases from the biomaRt central portal (v0.6) (https://www.ensembl.org/biomart/)[13] and TxDb.Hsapiens.UCSC.hg19.knownGene Annotation package. We selected functional genomic features in monocyte cells using the ChIP-seq (TF ChIP-seq) profile of K562 cells[110]. The first step in SNP enrichment analysis

was to choose g/tSNPs as foreground for each gene/transcript and background SNP sets from the 1 Mb window surrounding the TSS. We identified g/tSNPs with FDR less than 0.001 for the foreground (f), and other SNPs in the 1 Mb window around the TSS as background (b) SNP sets for each gene/transcript. We used the number of overlaps between foreground (f) and background (b) SNP sets in the genomic feature (chromatin state) and calculated the z-score (Z) of enrichment as follows:

$$Z = \frac{f/F - b/B}{SE(f/F - b/B)} \tag{1}$$

where $SE_{P1-P2}$ is the standard error of the difference between proportions. The method is implemented as an R package, called FEVV (Functional Enrichment of Genomic Variants and Variations). FEVV is available at: https://github.com/isarnassiri/FEVV/.

**Functional significance score of SNPs.** 3DSNP score was applied to evaluate the functional significance of an indicated SNP in 6 categories, including 3D interacting genes, enhancer state, promoter state, transcription factor binding sites, sequence motifs altered, and conservation categories[111]. The score for an SNP is calculated using the number of hits in each functional category in human monocytes from the GTEx project[112] and a Poisson distribution model[113].

**Inferring direction of the causal relationship between DNA methylation, genetic and expression.** We apply the MEAL R package to compute a Pearson correlation test between the methylation and gene expression values. The g/tQTL-mQTL pairs with shared causal variants and a significant correlation between methylation and gene expression values (FDR < $1 \times 10^{-4}$) in naïve and LPS-stimulated monocytes were selected for downstream analyses.

We used colocalization tests of two genetic traits (coloc)[106] with default parameters to assess whether g/tQTL and mQTL association signals are consistent with a shared causal variant. The posterior probability (PP) is the probability that there is a link between gene expression and methylation, both of which are associated with the SNP. It is assumed

that family members or related individuals were not included (abf: all but family). Coloc examines the posterior probability of four hypotheses (H), including PP.H1.abf (SNP associated with gene expression only), PP.H2.abf (SNP associated with methylation only), PP.H3.abf (SNP associated with neither trait), and PP.H4.abf (SNP is associated with both gene expression and methylation). The SNP's high value of PP.H4.abf suggests that it could have a causal effect on both gene expression and methylation, indicating a potential regulatory relationship[106]. To identify causal relationships in the region of interest, we utilised the PPH4 > 0.8[106] and the Steiger statistical test to analyse the deviation from independence between the direction of causal effects ($p < 0.05$)[114].

**Comparison and replication of gQTL results.** We compared gQTL profiles of LPS or IFN-γ treated primary monocyte cells formed on microarray profiling[3] with the gQTL formed on RNA-seq profiling of gene expression ($P < 0.01$). The experimental setup and microarray transcriptomic data of 367 individuals after exposure to IFN-γ, 322 individuals after 24 h LPS and 414 individuals in the naïve state were presented in detail in our previous study[3]. The replications were examined using an exact match of SNP-gene pairs from significant gQTL profiles. We use the qvalue method to calculate $\pi_1$ statistics in order to estimate the expected true positive rate[115]. The proportion of false positives ($\pi_0$) is calculated by assuming a uniform null P value distribution, and $\pi_1$ is equal to $1 - \pi_0$ [116].

We further validated our bulk RNA-seq gQTL analysis by conducting a comparison study with published single-cell RNA-seq (scRNA-seq) data[7]. The scRNA-seq dataset included monocyte cells from 120 individuals that stimulated or were induced in vitro with Pseudomonas aeruginosa (PA24). The objective was to determine if the gQTL associations that were identified were strong enough to replicate at the single-cell level. By comparing SNP-gene pairs and their gQTL p-values, we were able to study the replication of the scRNA-seq dataset to the RNA-seq dataset.

**Genetic determinant of gene regulatory network relationships.** We performed an allele-specific co-expression analysis[15,16] searching for variants that impact co-expression relationships of genes in cis-gQTL with upstream pathway genes.

First, we selected independent gQTLs (FDR < $1 \times 10^{-5}$, MAF > 0.039) with the homozygous minor allele count of > 5. Next, for each set of individuals with the same genotype, correlation coefficients ($r$) were calculated for expression values of the eGenes and all other genes. We used correlation coefficients and performed the genotype-specific differential gene correlation analysis (DCA) of samples with MM (M: minor allele) genotype versus samples with RR (R: reference allele) genotype, and samples with MM genotype versus samples with RM genotype as follows.

Fisher z-transformation was applied to stabilise the variance of rank correlation coefficients ($r_{MM}.r_{MR}.r_{RR}$)[46]:

$$z = \frac{1}{2} log_e \left( \frac{1+r}{1-r} \right) \qquad (2)$$

The difference in z-scores ($dz$) between two genotypes (e.g. $r_{MM}$ and $r_{MR}$) was calculated by:

$$dz = \frac{(z_1 - z_2)}{\sqrt{|var(r_{s_1}) - var(r_{s_2})|}} \qquad (3)$$

where $var(r_{s_x})$ refers to the variance of $z$ for the set of individuals with the same genotype ($s_x$). We can summarise the differential correlation relationships (DC) as the following logical statement:

$$p \bigwedge q \therefore d \equiv p \bigwedge q \therefore \neg d \qquad (4)$$

where $p : r_{RR} \& r_{MM} \in DC$, $q : r_{RM} \& r_{MM} \in DC$, and $d : r_{RM} \& r_{RR} \in DC$.

DCA uses the direction of correlation to categorise the relationship between gene pairs across different allele carriage at specific gQTL. DCA considers the significance and direction of correlation in each condition beyond the binary gain and loss of correlation, the relationships being divided into three categories: significant positive correlation, no significant correlation, and significant negative correlation. When these categories are combined across genotypes, there are a total of $3 \times 3 = 9$ possible differential correlation classes.

Using independent microarray-based analysis, we attempted to replicate coExQTLs. Those coExQTLs that had the same gene pair, SNP, and direction of correlation in both datasets were defined as replicated. The RNA-seq and microarray techniques have inherent differences that make it challenging to replicate coExQTLs findings using microarray data. For instance, RNA-seq has a much greater dynamic range of sensitivity to expression than microarrays. We used a conservative threshold to define the replicated coExQTLs in the replication study, and we believe we underestimate the number of coExQTLs in this first description.

We combined gene expression, CpG site methylation level, and genotype data to identify coExQTLs that regulate gene expression and DNA methylation. Our approach involved overlaying coExQTLs and methylation levels onto the co-expression networks in order to identify coExQTLs that control both gene expression and DNA methylation within these modules.

A more refined and biologically relevant understanding of gene regulation and its implications for complex traits can be gained by using transcripts instead of gene-level coExQTL. To identify genetic variants that are associated with transcript expression, we performed coExQTLs using transcript and gene (canonical transcript) expression profiles. The expression levels of transcripts (FPKM) were transformed using SAVER method[117]. The SAVER method was our preference because the benchmark studies indicated that it effectively establishes the correlation between marker genes and excludes those we are aware do not correlate[117].

The Benjamini-Hochberg (BH) correction method was used to adjust the resulting p-values. To find a balance between sensitivity and specificity, we only analysed transcript-level coExQTLs for independent-g/tQTLs with FDR less than $10^{-5}$ and MAF > 0.039. By comparing coExQTLs with $P_{interaction} < 10^{-5}$ in each condition to coExQTLs with $P_{interaction} > 0.01$ in other conditions, we determined context-specific coExQTLs for every condition.

We utilised IsoVis[118] and 3DSNP (version 2.0)[111] to perform functional enrichment analysis for annotating the identified coExQTLs. To find a balance between sensitivity and specificity, we only analysed transcript-level coExQTLs for independent-tQTLs with FDR less than $10^{-5}$ and MAF > 0.039. Independent-tQTL is a term used to describe a significant connection between an SNP and a transcript expression level that persists after conditioning on other SNPs in the same genomic region. Conditional pass analysis was utilised to detect independent-g/tQTLs[28].

**Pathway-based differential co-expression test.** A coExQTL's functional impact would be vary depending on the specific context and the biological significance of the gene within the pathway. A pathway-based differential co-expression (PDC) analysis was employed to determine whether a coExQTL is functionally neutral or impactful within a pathway[45,46]. PDC analysis was carried out on a specific group of genes that have genotype co-expression relationships with a specific coExQTL and were enriched with curated gene sets from online pathway databases.

The PDC analysis uncovers the average shift in correlation between gene expression among two genotype classes within a pathway, as well as its statistical significance. PDC calculates a differential connectivity score for every gene within each module. The score reflects the change

in the gene's connectivity within the module between the two genotype classes. When calculating the overall module differential connectivity, only genes that exhibit a significant change in connectivity ($p$-value < 0.05) are taken into account. Let $z_1$ be the z-score of gene $i$ and $j$ in genotype class 1, $z_2$ be the z-score of gene $i$ and $j$ in genotype class 2, and $n$ be the total number of genes in an indicated pathway. Then, the median difference in z-scores between all gene pairs (overall module differential connectivity) can be calculated as[45,46]:

$$\left(1 - \sum_p^n median\left(\left|dz_{i,j}^p = \frac{(z_1 - z_2)}{\sqrt{\left|var(r_{s1}) - var(r_{s2})\right|}}\right| for\ all\ i, j\ where\ i \neq j\ and\ 1 \leq i, j \leq n\right)\right)/n \quad (5)$$

The median function determines the middle value among all pairwise differences between two genotype classes, $p$ is interpreted as a permutation to compute a two-sided p-value and $var(r_{s_x})$ refers to the variance of $z$ for the set of individuals with the same genotype ($s_x$). The median estimator is our preferred choice due to its higher breakdown point compared to the mean. Spearman correlation is our preferred method for PDC due to its ability to handle non-linear monotonic relationships and non-normal distributions, which are common characteristics of gene expression data. Furthermore, Spearman correlation is less affected by outliers, which can occur in gene expression data because of technical noise or biological variability. The use of this approach has been previously introduced and implemented[45,46,119].

To determine the empirical p-value related to the observed effect, a permutation test with 1000 resampling was performed[45,46]. The threshold for gene expression variation within a pathway that would make a coExQTL functionally disruptive is 0.05.

The clusterProfiler tool was utilised to examine pathway enrichment of allele-specific co-expression relationships and display functional profiles[120]. We used hc2.all.v2024 as a reference gene set. hc2.all.v2024 is a complete gene set collection that has been created specifically for human genes and encompasses a wide variety of pathways from various biological processes[121]. The background population is the entire set of genes being considered in the analysis.

### Data presentation
Manhattan plots of genomic analysis were generated using CMplot (v3.3.3). Local association plots were generated with FUMA (v1.3.5)[122]. GWAS4D was used to integrate transcription factor binding sequence motifs with context-specific regulatory variants and visualise motifs as a WebLopo plot[123]. Box plots and dotplots were generated using ggpubr (v0.2) and customising ggplot2[124]. Visualisation of SNPs and methylation data along with annotation as track layers (Lolli plot) was generated using trackViewer (v1.20.2)[125]. We used shinyCircos (V2.0) to visualise g/tQTL association as a Circos plot[126,127]. The ChIPseeker package (v1.18.0) was applied to visualise feature distribution and distribution of eSNPs relative to TSS[128]. Alternative gene isoforms were visualised and annotations using the IsoVis webserver[118]. We leveraged Shiny with R to develop a web application framework on g/tQTL data for programming-free graphical and interactive analysis. The DBI R package was used to execute SQL queries and assign the results as the input of Shiny (https://livedataoxford.shinyapps.io/fairfaxlab_supplementary_files/).

### Reporting summary
Further information on research design is available in the Nature Portfolio Reporting Summary linked to this article.

## Data availability
We have made a browser available for independent g/tQTL at https://livedataoxford.shinyapps.io/fairfaxlab_supplementary_files/. The simplified version of the Shiny app is available online on shinyapps.io: gQTLs: https://livedataoxford.shinyapps.io/fairfaxlab_monocytes_eqtl_lps/

https://livedataoxford.shinyapps.io/fairfaxlab_monocytes_eqtl_ifn/ tQTLs: https://livedataoxford.shinyapps.io/fairfaxlab_monocytes_tqtl_lps/ https://livedataoxford.shinyapps.io/fairfaxlab_monocytes_tqtl_ifn/ All sequencing data are made freely available to organisations and researchers to conduct research following the UK Policy Framework for Health and Social Care Research via a data access agreement. Sequence data have been deposited at the European Genome–Phenome Archive, which is hosted by the European Bioinformatics Institute and the Centre for Genomic Regulation under accession no. EGAS00001007111 [https://ega-archive.org/datasets/EGAD00001010176].

## Code availability
Scripts used in the analysis and figure synthesis are available online on shinyapps.io: https://livedataoxford.shinyapps.io/fairfaxlab_supplementary_files/.

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

## Acknowledgements

The research was supported by the Wellcome Trust Core Award Grant Numbers 203141/Z/16/Z and 226535/Z/22/Z to B.P.F. J.C.K. is supported by a Wellcome Trust Investigator Award (204969/Z/16/Z), the National Institute for Health Research (NIHR) Oxford Biomedical Research Centre and the Chinese Academy of Medical Sciences (CAMS) Innovation Fund for Medical Science (grant 2018-I2M-2-002), Wellcome Trust grants 090532/Z/09/Z and 203141/Z/16/Z to the core facilities of the Wellcome Centre for Human Genetics was supported by I.N. was supported by the National Institute for Health Research (NIHR) Oxford Health Biomedical Research Centre (BRC). We are grateful to the Oxford Biobank for providing access to the samples utilised in this study. We are thankful to the Oxford Biobank participants for their willingness to participate in medical research. The views expressed are those of the author(s) and not necessarily those of the NHS, the NIHR or the Department of Health. The graphics used in Fig. 1 were partly adapted from Servier Medical Art (https://smart.servier.com/), licensed under CC BY 4.0 (https://creativecommons.org/licenses/by/4.0/).

## Author contributions

The study was conceived by B.P.F. who oversaw the project. Samples were collected by S.D., E.L., B.P.F., H.A.M. and J.J.G. with access to the biobank provided by MN. Primary analysis was performed by IN, who also developed website browsers of data, with input from J.J.G., O.T. and BPF. The manuscript was draughted by IN and BPF with input and revisions from all other authors.

## Competing interests

The authors declare no competing interests.
