## [Transparent Peer Review file · Nature Communications]

Genetic determinants of monocyte splicing are enriched for disease susceptibility loci

Corresponding Author: Professor Benjamin Fairfax

Version 0:

Reviewer comments:

Reviewer #1

(Remarks to the Author)

This manuscript authored by Nassiri et al. investigates the correlation between SNP genotypes and both array-based DNA methylation and RNA-seq derived gene and transcript expression levels in monocytes. These were measured in both resting states and after 24-hour stimulation with LPS or IFN-gamma, utilizing a cohort of 139 or 174 samples. The study identifies statistically significant cis-mQTL, gQTL, and tQTL across both conditions.

Additionally, the authors examined the top cis-gQTL for deviation from co-expression with other genes in the same pathway, and conducted co-localization tests to determine overlaps with significant disease-associated GWAS variants. The reproducibility of the identified gQTL was compared with previous array-based studies, showing a replication rate of 12-75%.

The authors provide a publicly accessible web-based browser for data querying, along with the availability of summary statistics and analysis scripts.

The manuscript presents a comprehensive dataset of QTL in monocytes subjected to immunologically relevant stimuli, with a particular emphasis on RNA-seq based splicing QTL under such conditions. By focusing on monocytes, the study circumvents confounding factors arising from cell type composition, thereby enhancing the interpretability and precision of the findings. The discovery of GWAS co-localization signals linked to disease phenotypes generates promising hypotheses for subsequent targeted experimental investigations.

While the research is fundamentally robust, it would benefit from an improved presentation, refinement in methodology, and a more nuanced interpretation of mQTL results.

Specific Comments:

1. The manuscript appears to omit citations of pertinent literature that could contextualize the current findings (e.g., PMID: 28814792, PMID: 23382694, PMID: 24786080). It is recommended that these works be discussed in relation to this study.
2. There is a discrepancy in the sample size reported in the abstract (n=185) and the Methods section (n=174 or n=139). This inconsistency should be addressed for clarity.
3. The presence of first or second-degree relatives within the dataset could introduce bias. The authors should clarify if such relationships exist and describe how the analytical models account for relatedness.
4. The use of Illumina 450K array probes that contain SNPs may confound methylation results (PMID: 26607064). It is advisable to either exclude these probes or adjust for their presence in the analysis.
5. Consideration of sex and age as variables in the analysis is crucial, given their known effects on immune responses. The identification of sex-specific stimulus QTLs would be an intriguing aspect to explore (refer to PMID: 38072919 and PMID: 31024623 for further insight).
6. The manuscript reports differential regulation of mtDNA by ERAP2 in monocytes, depending on the rs2910792 genotype.

It would be beneficial for the authors to discuss whether this phenomenon is unique to monocytes or observed in other immune cells. Additionally, elucidation of the functional implications of increased mitochondrial count in monocytes and its physiological relevance would enhance the understanding of this finding.

7. The manuscript would also benefit from an expanded discussion on the potential confounding effects of mtDNA count variability due to age, disease states, and oxidative stress. Specifically, the implications of these factors on the study's findings, such as the association between rs2910792 and ERAP2, should be addressed. Furthermore, the authors should critically evaluate the representativeness of their sample set concerning the patient cohorts in the cited GWA studies (such as IBD) where rs2910792 co-loc was detected.

8. The threshold for gene expression variation within a pathway that would render a QTL functionally disruptive is not clearly defined. Without this information, it is challenging to determine whether a QTL is functionally neutral or impactful. To ascertain the functional significance of identified coExQTL, clarification is needed regarding the permissible range of gene-specific expression variation within a pathway that would categorize a QTL as functionally disruptive.

9. Section 2.2 lacks clarity on the application of conditional analysis for the analysis of tQTL. A more detailed explanation of the conditioning process within the analyses is necessary.

10. Given the minor allele frequency (MAF) threshold of 0.04 and the sample sizes ($n=174/139$), the study may be underpowered for detecting effects associated with many of the included SNPs. A power analysis at the selected false discovery rate (FDR) level is recommended, with subsequent exclusion of potentially low-powered QTL.

11. The method utilized for calculating the FDR is not specified. Given that the multiple p-values corrected for FDR are likely correlated (e.g., due to local CpG density affecting methylation, linkage disequilibrium among SNPs, interconnected genes within pathways, etc), an FDR calculation method that accounts for these dependencies is necessary.

12. There is a concern regarding unaccounted confounders or model misspecification leading to p-value inflation. The authors could estimate this inflation by presenting QTL p-value distributions as quantile-quantile (QQ) plots and calculating genomic inflation (λ) values. Should inflation be detected, a re-evaluation of the models is warranted.

13. The assertion that a relationship between cg22687766 methylation and CD55 gene expression is established post-LPS is premature without causality analysis or specific supporting data. Figures 3d and e, as well as Supplementary Figure 7, should include causality analysis or additional data to substantiate the proposed interaction between DNA methylation and gene expression. As it stands now, the relationship between methylation and CD55 expression can be explained by rs2914937 genotype that affects both, thus being a confounder.

14. The manuscript would be more accessible to readers if it included schematic diagrams to illustrate the study design and concepts.

15. There are inconsistencies in terminology and acronym usage throughout the manuscript. For example, the acronym "PP.H4" is not defined, and there are variations in the terms "QTL" vs. "QTLs," "IFN- γ " vs. "IFN γ ," and in 2.4.1 an error in "casual" instead of "causal".

16. The authors should complete the legend for Supplementary Figure 1 by providing descriptions for panels a) and b). Furthermore, an explanation of the purpose and selection criteria for the 0-50 PCs tested in gene expression profiles is necessary.

17. The font size in the main figures is occasionally too small for comfortable reading, and the contrast of the text in panel strips is too low to be readable. Figure 4 resolution is significantly lower than in others. To enhance readability, the font size in the main figures should be increased to at least 5pt.

(Remarks on code availability)

Reviewer #2

(Remarks to the Author)

Nassiri et al. have conducted an extensive bulk QTL study in immune-stimulated monocytes of 185 European donors. More specifically, they assessed methylation QTLs, both transcript- and gene-level eQTLs, and co-expression QTLs. Using linear regression and co-localization analysis, they found that many ($\geq 86\%$) epigenetic and expression-level changes share a likely causal variant. Finally, they co-localized the context-specific transcript- and gene-level eQTLs with GWAS risk loci and noticed that depending on the context different diseases are co-localizing with the eQTL signal.

While the analyses have been conducted with care, currently, the novelty of the study is more limited. Many context-specific eQTL (e.g. Fairfax et al. 2014; Kimm-Hellmuth et al. 2017) and (less so) co-expression QTL analyses (e.g. Oelen et al. 2022) have already been conducted in the past – also in monocytes, and by the authors themselves. Moreover, many of these efforts have been previously uniformly processed and combined in the eQTL catalogue, a compendium of gene expression, exon expression, transcript usage and splicing QTLs (Kerimov et al. 2021). The main novelty could have been combining

the transcript-level eQTLs with the co-expression QTL analyses, however, this analysis was not conducted.

Major comments:

1. To increase the novelty of this study and make the manuscript more of a complete story, it would be very interesting to conduct the co-expression QTL analyses using the transcript eQTLs instead of the gene eQTLs. Currently, the manuscript feels like a lot of independent analyses have been put together without a clear storyline that connects them, this would be one way to make it a more coherent/connected story.

Previously, it was shown that genetic variation can decouple/affect the upstream regulatory network of eQTL genes – at the gene eQTL level (e.g. Van der Wijst et al. 2018; Oelen et al. 2022). By now conducting co-expression QTL analyses using the transcript eQTLs, you may find how different isoforms have different upstream regulators that may also provide an explanation for some of the isoform switching that you observed (e.g. OAS1, Fig. 2e).

2. Should the focus of the manuscript not be completely revised? Isn't the main message that tQTLs are much more informative than gQTLs? As they are more enriched for disease associations, may overlap better with mQTLs (described in methods section that this was done, but it's not discussed in the results section at all – why?), novelty in transcript QTL co-expression QTL analysis, etc?

3. The paper would greatly benefit from copy editing, as several sentences are way too long or are sloppy due to missing words or just generally very difficult to comprehend. As a result, the sentences are sometimes not understandable at all. To give a few examples of sentences that are way too long, miss words or that I did not understand at all:

a. 'Correspondingly, a significant proportion of monocyte eQTLs? are only apparent upon monocyte activation 3, 6.'

b. 'Further exemplar tQTLs demonstrating high degrees of conditional specificity include to the genes RGS1, DDX1, CTSC and KIFC3 (Supplementary Figures 4 and 5), as well as tQTL to genes where the same SNP influences tQTL formation in opposing directions to different transcripts according to immune state for example, MICA (Supplementary Figure 6).'

c. In untreated monocytes, this locus demonstrates weak gQTL activity (REPORT EQTL Z-SCORE INSTEAD?) (OAS1 FDR=0.0027, OAS3 FDR=0.0023), however expression of both these genes is robustly upregulated by exposure to IFNg and this state correspondingly associated markedly increased significance of these associations (OAS1 FDR = 10-14; OAS3 FDR = 3-21) (Figure 2d).

d. One example of such an effect where context leads to establishment of a genetic relationship is at rs2914937 (chr1:207315423G>A) which is an mQTL for cg22687766 (PP.H4 = 1) within the upstream promoter (FDR 10–58) across both resting and LPS treated states, and specific to post LPS state forms an gQTL for CD55 (FDR 10–63), and an ??? (Figure 3c,d).

e. Genetic determinants of condition-specific expressed transcripts

This paragraph seems to be in a random position in the supplementary information. This part especially was badly written, and I had no clue what the entire paragraph was doing here.

4. To further enhance the coherence of the story: When you discuss the gQTL and tQTL results (in the 'Identification of cis-acting QTL' section), can you also mention something about the overlap between gene- and transcript-level results? So how many gQTLs are also a tQTL, how often are they concordant, how often are they condition-specific, etc?

We found that 54.6% of tQTL (4763 observed over 8727 total), showed context specificity, of which 3.4% involve more than one transcript associating with the same regulatory variant. e.g. are these tQTLs with one variant regulating multiple transcripts from the same gene more often a gQTL?

These types of analyses will give the reader a bit more information about how relevant it is to also look for eQTLs at the transcript-level instead of the gene-level (the level at which most studies are currently being conducted), and will give some more molecular insights into the underlying processes.

5. To further enhance the coherence of the story: Why not integrate the DNA methylation information for the interpretation of the co-eqtls? Does it explain the context-specificity?

6. Using a complementary Bayesian approach (Moloc)19, 20, we identified evidence (PP > 0.5) supporting 3572 gQTL and 2016 tQTL being unique to one condition only (Supplementary Table 2).

Could you provide more clearly what the complementarity is here? First, it is not made explicit to what exactly it is complementary. Second, it is not made very explicit what you assess with Moloc. Could you elaborate a bit more so that it is also clear for someone who doesn't have heard from Moloc before?

7. Using Mendelian randomization, we found that tQTLs are enriched for loci overlapping disease processes (FDR < 1-10 and PP.H4 > 0.8), with tQTL from the untreated monocytes showing marked enrichment for rheumatoid arthritis and cancer, and LPS- and IFNg-stimulated monocytes showing enrichment for asthma (Figure 2a).

'disease processes' = GWAS for disease?

Enriched compared to what? And more enriched than gQTLs? In other words, is it important to look at tQTLs in addition to gQTLs? Or what is the message that you want to convey here?

8. For certain traits, disease enrichment using Mendelian randomisation (MR) was most significant for gQTL post treatment, such as MS and LPS associated gQTL, whereas for inflammatory bowel disease and COVID19 risk-loci, enrichment was most marked in IFNg induced gQTL (Figure 2a-b and Supplementary Table 3).

This part that follows is suddenly about gQTLs again. There is no connection in the storyline at all, it feels very disconnected. As mentioned in the comment above, by discussing the results in comparison (tQTL vs gQTL) you create much more of a connected storyline.

Dont you also wanna explain whether these findings are as expected or not? So are the diseases you mention known to be IFN-induced or association LPS and MS... Does that make sense?

9. The reproducibility of the cis-gQTLs has been assessed against micro-array-based results from an earlier cohort. It would be also very interesting to see how many of the cis-gQTLs can be reproduced using pathogen-stimulated monocytes in scRNA-seq data (Oelen et al. 2022). Although the stimulations are slightly different in this study, they are for the same duration (24h) and many of the same biological processes will be activated by these pathogens. An important difference is that this study uses pathogen-stimulated PBMCs instead of isolated monocytes. So potentially some biological processes will not be activated in monocytes that are stimulated without the other immune cells present. Including such a comparison

gives valuable, more general insights into the reproducibility between single-cell and bulk QTL studies, and which QTLs might be missed in an (even) more artificial system of isolated monocytes stimulated with a single cytokine (IFN γ) or pathogen (LPS) component.

10. Co-expression QTL mapping: although the methods section describes co-expression QTL analyses conducted in all three genotype groups, the results section only reports the analyses conducted between homozygous ref and alt groups. Moreover, these results are not placed into the context of other similar publications: in bulk, e.g. Zhernakova et al 2017 describes the interaction between genetic variants and pairs of genes in interaction analyses, and in single-cell, e.g. Wijst et al 2018 and Oelen et al 2022 have leveraged single-cell measurements to calculate correlations per individual. Importantly, bulk as opposed to single-cell co-expression QTL analyses have the risk to identify false correlations due to Simpsons paradox (Li et al 2023), meaning that in bulk measurements the actual direction of effect may be masked as co-expression correlations are measured across donors instead of per donor. Therefore, replication of co-expression QTLs in a single-cell study is preferred (e.g. Oelen et al. 2022) over the replication in the microarray study.

11. Using a conservative approach and only described coExQTLs that demonstrate the same directional allele-specific differential gene correlation analysis (FDR<0.01) for the same gene pair, SNP, and direction of correlation across both datasets.

+

We observed 1576 genes to correlate with ERAP2 expression in a genotype specific manner, of which 764 (48.4%) replicated in the same genotypically divergent direction, the most significant being EPM2AIP1 (Figure 4h).

Could you report which proportion did not replicate in the micro-array study?

Could you report the concordance (same direction of effect) of all significant effects from the bulk RNA-seq study, in the micro-array study?

Again, replication in a single-cell study is preferred to overcome the limitations of bulk studies for co-expression QTL mapping.

12. Pathway analysis of co-expression QTL genes: did you use a background gene set, and if so, which one? In the methods section you describe a differential gene expression analysis but it is not clear where in the main text this is described.

13. Despite many efforts to construct personalized and context-specific gene regulatory networks^{18, 47, 48}, there has not been a systematic effort to apply differential gene correlation analysis in an allele-specific manner.

This is incorrect: see for example Zhernakova et al. 2017, Wijst et al. 2018, Oelen et al. 2022, Li et al. 2023.

14. There is already quite some literature on the role of OAS1 in COVID-19 (e.g. <https://www.nature.com/articles/s41588-021-00996-8> and <https://www.ncbi.nlm.nih.gov/pmc/articles/PMC8288155/>), but the findings ('notable disease relevant associations (PP.H4>0.99) include IFN γ specific tQTL at COVID-19 severity locus rs10735079, where allelic variation modulates context-specific splicing of OAS1, ...') are not really discussed in light of what has been already reported.

Minor comments:

1. Can you briefly explain in the results section why you have chosen different windows for the gene vs transcript eQTLs? This helps the reader to better understand why things are being done the way they are presented.

2. Throughout the manuscript you use gQTL, tQTL and eQTL. You would expect that eQTL refers to both gQTL and tQTL here, but it seems to only refer to gQTL. Shouldn't you just keep using gQTL and tQTL throughout the manuscript to prevent any confusion?

3. It is unclear whether this difference in window size may have impacted your results?

a. Whereas the median distance for peak gQTL from the transcription start site (TSS) was 14648 bp (95% CI [13982 – 15435]), peak tQTL are typically more proximal to TSS (median 5822 bp 95% CI [4938 – 6598] (UT), 6350 bp 95% CI [5176 – 7362] (LPS), 6248 bp 95% CI [4901 – 7688] (IFN γ)).

15. Did you limit your analyses only to the 100 kB window size for both the tQTLs and gQTLs? If not, you should confirm that the observed differences between tQTLs and gQTLs are not a consequence of the different window sizes (1Mb for gQTL vs 100 kB for tQTL) that were used for the discovery.

16. For the interpretation / comparison of tQTL and gQTL results in Figure 1D it is relevant to mention in the legend that a different mapping window is used, as this may affect the relative distribution to which genomic feature a QTL is assigned to. So does this imply that gQTL and tQTL cannot be fairly compared in Figure 1D and 1E and should be presented in separate figures? Or does it mean that any conclusions drawn from here that are cross gQTL and tQTL should be confirmed in a 100 MB window gQTL mapping?

4. Fig. 1D-1E: Similarly, the number of gQTL and tQTL within a specific genomic feature or chromatin state may vary, even independent of the chosen window size. As a result, the gQTL and tQTL cannot be fairly compared within the same feature or state. So does this again imply that gQTL and tQTL cannot be fairly compared in Figure 1D and 1E and should be presented in separate figures?

5. Please write out abbreviations in full the first time they are being used, e.g. 'TxFlnc marks'.

6. All p-values/numbers in the manuscript (main text and figures) are weirdly presented and don't seem to reflect the actual accuracy of the presented value: e.g. not 1-14 but 1.00×10^{-13}

7. For all figures: when the QTL effect is not significant, 'na' is reported. Could you give the actual p-value instead, and highlight significance for example using asterisks? Additionally, a rho, Z-score or slope should be given for all the QTL plots.

8. For all figures: could you mention in the figure legend what the error bars represent and the N for each figure is? Could you also mention in which measurement the Y-axis is displayed (Fig. 1F: e.g. 'Gene expression' in FPKM?)

9. For all figures: readability should be improved by increasing the font size of all axes and numbers displayed.

10. rs4072037 allele carriage unclear: you should specify which SNP allele you mean (the risk allele apparently after reading the rest of the sentence, so which allele is that?)

11. but which notably had no gQTL post IFN γ 24 (Figure 1g). do you mean that it doesn't have a gQTL but did have a tQTL? Or do you mean it has no gQTL after IFN γ stimulation but did have a gQTL in unstimulated? The way you write it, it seems to imply the latter, but I think you mean the first. Better to rewrite to make this more clear, e.g. 'We found IFN γ exposure induced

tQTL (but not gQTL) ...'

12. Fig. 1A-1C: it's unclear why specific eQTL examples are highlighted. Would be good to clarify this in the figure legend.
13. Fig. 1D: (d) Functional consequences of lead tSNPs on genomic features. 'functional consequence' should be rewritten, e.g. as 'functional annotation of the genomic feature wherein the lead tSNP resides', as you don't assess the consequence of the SNP but just annotate the genomic feature in which this SNP is located, right? in comparison with background SNP. How is the background SNP defined here?
14. Fig. 1E: peripheral blood (E029) unclear what this means or refers to.
15. Fig. 1F: The arrows in fig. 1F are distracting from the point you want to make as they are missing in some introns and don't highlight the actual differences among the transcript variants. It's maybe also good to highlight where the differences in transcript isoforms are in the two highlighted isoforms? Because apparently that's the region that explains the different downstream consequences! Currently, it's almost like a find-the-differences puzzle.
16. The abbreviations that are used in Fig. 1E and 1F are unclear and therefore should be explained in the figure legends.
17. Fig. 2B: The figure seems to imply that disease co-localization of tQTLs is better than for gQTLs, and that the naive condition shows highest co-localization. However, the amount of gQTLs and tQTLs differ already widely per condition, so shouldn't it be more informative to show the proportion of QTLs that co-localize as well? Additionally, to show that the context-specificity is so important for capturing disease-associated effects, could you also show an extra bar with those tQTL or gQTL that are independent of condition?
18. Fig. 1F and 2C: It's confusing that there is more isoforms with IFNg vs LPS (in isoform usage figures) - why is that? If there's neglectable expression, it may still be good to keep all isoforms in there, so it's clear? And why does this even need 2 separate figures (both 1F and 2C)? Isn't it very easy to put all information in one figure? Now a lot of data is repeated twice (UT) and it doesn't make it easier to compare when they are in different plots.
Fig 2C  x-axis label 'LPS24' seems incorrect. Y-axis label can be more clear  why not specify that it's 'OAS1 expression level' or 'OAS1 isoform expression level'?
19. Fig. 2D-2F: the figures are not logically ordered. Even though the same SNP is affecting both OAS1 and OAS3 gQTL effects, the OAS3 gene does not come back in Fig. 2C or 2E, so it's distracting/confusing. I would suggest to 1. remove OAS3 and IFNAR2 all together, or 2. to group OAS1, OAS3 and IFNAR2 gQTLs as one figure being the 'COVID19 risk loci associated gQTLs', and position it before fig. 2C. Old Fig. 2C and 2E can then follow as an example of such a risk locus that is further explained by tQTL effects, but maybe it's good to then highlight that no tQTLs were observed for OAS3?
20. Figure 2C and 4J are not mentioned in the main text. Please make sure that all figures are referenced and discussed, in the order they appear in the figures.
21. The differential splicing of ENST00000452357 skips an exon in ENST00000202917 with rs10735079 (chr12:112942203G>A) it would be good to highlight in Fig. 2C which exon is skipped, so that the figure is more informative and directive for the reader where to look at.
22. Fig. 3A: Figure looks beautiful, but I'm not sure what message you want to bring with this figure? Just a look-up table might be way more practical - and if the message isn't clear from this figure anyways then maybe removing it all together is better (or improving the text and legend to make the message clear)?
23. Fig. 3B: again maybe not the best way of visualizing the main message. Don't you want to know the relative proportion of each category within the positive vs negative effects (so scale differently)?
24. The departure from independence is coloured according to whether the residuals are positive or large. 'or large': no clue what you mean by this?
25. Fig. 3E: The adjusted p-value for the beta coefficient correlation between methylation and expression is 4.56×10^{-6} .
17. This p-value is independent of the genotype, or was it written unclearly in the legend? Additionally, could you explicitly test for a genotype x condition (unstim vs LPS) interaction effect instead, and report that p-value in Fig. 3E? And don't forget to report the rho of the regression lines and the non-significant p-values.
26. We confined our analysis to lead-gQTLs with $FDR < 1 \times 10^{-5}$ and $MAF > 0.039$.
18. Can you explain in the text why you have made these decisions (the numbers seem arbitrary at first glance)?
27. Correlation coefficients (r) were computed for the expression values of eGenes and all other genes for individuals homozygous for either allele...
19. Can you also specify in the main text that this is the Spearman r?
28. Supplementary tables: can you provide an explanation for all the column names in the first tab of each table?
29. an allele that shows recent selective pressure, purportedly due to the Black Death³⁶
20. There has been quite some commotion in the field about the paper that is being cited here, see for example <https://www.ncbi.nlm.nih.gov/pmc/articles/PMC10055098/>. Please be careful with the specific citation, maybe better to remove this remark all together as the scientific community is not convinced about the statements made in the cited ref ³⁶.
30. Fig. 4B, 4E, 4H: You report on the adjusted p-value of the correlation difference between the two stimulation conditions (naive vs LPS). Can you better explain what exactly has been tested? Did you put the stimulation condition as interaction term?
31. Fig. 4I: unclear what the red vs blue color represents.
32. Post LPS, coExQTLs are present at rs3110426 and OXR1 expression, with rs3110426 possibly affecting the binding site of ZNF628 transcription activator³⁸.
Unclear how this reduced binding would then result in a negative co-expression relationship (instead of the absence of any co-expression relationship).
33. Whilst we believe this work has furthered our understanding of the impact of immune conditions on regulatory genetics in monocytes, using LPS or IFNg single treatment models inevitably will fail to recreate the complex cytokine milieu and cell-cell interactions that characterize the immune system in health and disease.
21. All these issues are largely mitigated using stimulated PBMC single-cell data (e.g. Oelen et al. 2022).
34. 192 healthy individuals recruited via the Oxford biobank reference to Oxford Biobank is missing, and ancestry is not reported.
35. The shiny app browser works well and makes it easy for others to take advantage of the (significant) results. Also putting

the non-significant eQTL summary stats somewhere would be appreciated by the community.

36. Section 2.2 methods: 'zero to 50 PCs ... 'dominant PC'...' what does dominant mean here, please elaborate?

37. Section 2.2.1 methods: Why did you not use coloc or fine-mapping for identification of the lead SNP?

38. Section 2.4.1 methods: Why did you not use Susie to directly incorporate LD information while colocating with GWAS signal instead of manually looking for LD proxy SNPs when the SNPs were not in the eQTL output itself already?

39. Fig. S2: the difference between S2A and S2B is unclear, no information is provided in the legend text. Figure S2C is very complicated. There is many t-statistics provided, and it's very much a puzzle to figure out what each of these represent. Please improve the figures and/or explain the figures better in the legend.

40. Multiple times throughout the figures/text 'LPS24' is mentioned, but it is unclear why it is not just LPS as in many other figures? It's always 24h LPS stimulation, right? So why not keep it consistent and refer to it as 'LPS'?

(Remarks on code availability)

The code is not well organized and the description of required software, runtime, ordering and inputs of scripts needs to be greatly improved. The readme at the root of the repository only states the title of article with no other instructions supplied. There is little in the way of describing what analyses are found in which scripts and folders. The comments in the files themselves are also very scarce, giving little explanation to where specific steps are taken, and for which reasons. A great deal more detail would be required for someone to feasibly replicate the results described in the paper.

Version 1:

Reviewer comments:

Reviewer #1

(Remarks to the Author)

The authors have addressed all my concerns.

(Remarks on code availability)

The scripts are now documented with comments regarding the general purpose and outputs of the script, although individual commands are not commented or explained much. Nevertheless, it should be possible to replicate the study since the analysis relies mostly on methods from various well-documented R libraries for each step, and the code segments are explained. However, I haven't tried to run the code since I lack suitable input data files.

Reviewer #2

(Remarks to the Author)

We are happy to see that the authors have significantly improved the content and readability of the manuscript. The story has become much more coherent, and the novelty has very much improved.

A few small remarks that have remained:

1. The font size of the figures is still too small to read (at least in the merged pdf that was provided for review).

2. There is still a few textual errors in the main text, so a final proof-read would do good.

3. Some text is still too complicated / unreadable: e.g.

'Here IFN-exposure was found to induce tQTL (but not gQTL) to alternative transcripts of MUC1 (Figure 2f) including ENST00000612778 (FDR 2.7x10⁻⁶) and ENST00000620103 (FDR 8.0x10⁻¹⁵) according to rs4072037 carriage, a risk allele (rs4072037:G) for oesophageal and gastric cancer, illustrating the disease informing potential of this analysis (Figure 2g).'

4. In the naïve state, 36% (179/497) of gQTL and 42.4% (366/862) of tQTL demonstrate dependent causal effects on methylation linked to expression or vice versa, with this proportion being 40.3% (147/365) of gQTL and 43.3% (256/590) of tQTL post-LPS.

 Could you split this directionality up by methylation  expression or expression  methylation? The same for the following sentence about positive vs negative relationship between methylation and expression.

5. It's difficult to assess whether the co-expression tQTLs are specific for one isoform, or entail potentially multiple isoforms from the same gene. Additionally, if different SNPs regulate different isoforms from the same gene, do they have similar or different upstream co-expression QTL genes that are in co-expression with these tQTL effects?

(Remarks on code availability)

The code and descriptions are much clearer now, thank you.

Version 2:

Reviewer comments:

Reviewer #2

(Remarks to the Author)

Thank you for resolving the last few points. All is readable, clear and well explained now.

(Remarks on code availability)

Ok

Responses to Reviewer's Comments

We appreciate very much the reviewers for the constructive comments, and the effort and time put into the review of the manuscript.

Each comment has been carefully considered point by point and responded to. The reviewer and changes in the revised manuscript are as follows.

REVIEWER COMMENTS

Reviewer #1

This manuscript authored by Nassiri et al. investigates the correlation between SNP genotypes and both array-based DNA methylation and RNA-seq derived gene and transcript expression levels in monocytes. These were measured in both resting states and after 24-hour stimulation with LPS or IFN-gamma, utilizing a cohort of 139 or 174 samples. The study identifies statistically significant cis-mQTL, gQTL, and tQTL across both conditions.

Additionally, the authors examined the top cis-gQTL for deviation from co-expression with other genes in the same pathway, and conducted co-localization tests to determine overlaps with significant disease-associated GWAS variants. The reproducibility of the identified gQTL was compared with previous array-based studies, showing a replication rate of 12-75%.

The authors provide a publicly accessible web-based browser for data querying, along with the availability of summary statistics and analysis scripts. The manuscript presents a comprehensive dataset of QTL in monocytes subjected to immunologically relevant stimuli, with a particular emphasis on RNA-seq based splicing QTL under such conditions. By focusing on monocytes, the study circumvents confounding factors arising from cell type composition, thereby enhancing the interpretability and precision of the findings. The discovery of GWAS co-localization signals linked to disease phenotypes generates promising hypotheses for subsequent targeted experimental investigations.

While the research is fundamentally robust, it would benefit from an improved presentation, refinement in methodology, and a more nuanced interpretation of mQTL results.

Specific Comments:

1. *The manuscript appears to omit citations of pertinent literature that could contextualize the current findings (e.g., PMID: 28814792, PMID: 23382694, PMID: 24786080). It is recommended that these works be discussed in relation to this study.*

Author response:

We thank the reviewer for their positive comments and suggested references. In our revised submission we have added the relevant literature mentioned by the reviewer as citations in the revised manuscript and compared the results (e.g., Page 11, lines 443).

2. *There is a discrepancy in the sample size reported in the abstract (n=185) and the Methods section (n=174 or n=139). This inconsistency should be addressed for clarity.*

Author response:

We apologise for any confusion here. Whilst Table 1 demonstrates an overview of donors (n=185 total), with 176 being untreated, 176 being LPS, and 139 being IFNg, we have also made this clearer by including these details in the abstract (Page 2, line 39).

3. *The presence of first or second-degree relatives within the dataset could introduce bias. The authors should clarify if such relationships exist and describe how the analytical models account for relatedness.*

Author response:

Blood samples were taken from individuals who were recruited by the Oxford Biobank. We conducted an identity-by-descent analysis using PLINK. The results demonstrate that there were no close degree relatives within the cohort with maximum PI_HAT <0.05 (median PI_HAT=0), corresponding to less related than cousin (PI_HAT=0.0625, third-degree). To clarify, we have added these details in the method section (Page 14, lines 560-562).

4. *The use of Illumina 450K array probes that contain SNPs may confound methylation results (PMID: 26607064). It is advisable to either exclude these probes or adjust for their presence in the analysis.*

Author response:

Thank you of your comment. We removed probes containing SNP(s) at/near the targeted CpG site, regenerated the figures accordingly, and added the following sentence and related citation to the method section (Page 13, lines 535-537):

“We excluded 96,427 loci that were analysed using probes that contained SNP(s) at/near the targeted CpG site (≤ 50 base pair), given their potential to introduce spurious associations.”

5. Consideration of sex and age as variables in the analysis is crucial, given their known effects on immune responses. The identification of sex-specific stimulus QTLs would be an intriguing aspect to explore (refer to PMID: 38072919 and PMID: 31024623 for further insight).

Author response:

The reviewer makes an interesting observation regarding sex-based eQTL. However, this was not the primary aim of our work and sex-based eQTL analyses can form the bases of entire publications. Further, we are concerned that the power of our sample size is insufficient for sex-specific stimulus eQTL analysis. We appreciate this is an area of interest however and will make all data accessible to permit other researchers to perform such analyses. A new paragraph was added to page 16 (lines 630-632) where we discuss this.

6. The manuscript reports differential regulation of mtDNA by ERAP2 in monocytes, depending on the rs2910792 genotype. It would be beneficial for the authors to discuss whether this phenomenon is unique to monocytes or observed in other immune cells. Additionally, elucidation of the functional implications of increased mitochondrial count in monocytes and its physiological relevance would enhance the understanding of this finding.

Author response:

We thank the reviewer for this point. We note that rs2910792 forms a pan cell-type eQTL, however expression levels of this gene appear maximal in immune cells with a role in antigen presentation (we have provided a Supplementary Figure 10 on page 60 demonstrating this).

Whilst the elucidation of functional implications would be interesting, it would require a specific physiological setup (use of Seahorse or similar equipment) which our group is not established to perform and is outside the scope of this genetic study. We have however addressed the functional effects of differential regulation of mtDNA by ERAP2 on monocytes and other immune cells (Page 10 lines 382-389) in the revised manuscript. Furthermore, we have included an extra figure to examine the level of ERAP2 expression in various human tissues, including immune cells (Supplementary figure 10 on page 63).

7. The manuscript would also benefit from an expanded discussion on the potential confounding effects of mtDNA count variability due to age, disease states, and oxidative stress. Specifically, the implications of these factors on the study's findings, such as the association between rs2910792 and ERAP2, should be addressed.

Furthermore, the authors should critically evaluate the representativeness of their sample set concerning the patient cohorts in the cited GWA studies (such as IBD) where rs2910792 co-loc was detected.

Author response:

We thank the reviewer for these points – our samples are from a healthy biobank (<https://www.oxfordbiobank.org.uk>) and as such – in keeping with other healthy control cohorts (including the UK biobank) will likely differ in terms of general health (cohort studies tending to have over-representation of those with improved sociodemographics). However, we do not find that there is an association between either age or sex and mitochondria count in these cells (as shown in the below Figures 1 and 2), thus any impact of sociodemographic would be expected to minimal compared to the profound effect of exposure to IFN-gamma and genetics. We have reflected on this point in a new paragraph has been added to the revised version of the manuscript to review the experiment and computational procedure we employed to provide mtDNA count (Page 12 lines 519-521 and page 9 lines 369-372).

Figure 1. Association between mitochondrial count and age.

Figure 2. Boxplot showing the distribution of mitochondrial count between males and females.

8. *The threshold for gene expression variation within a pathway that would render a QTL functionally disruptive is not clearly defined. Without this information, it is challenging to determine whether a QTL is functionally neutral or impactful. To ascertain the functional significance of identified coExQTL, clarification is needed regarding the permissible range of gene-specific expression variation within a pathway that would categorize a QTL as functionally disruptive.*

Author response:

Our description of coExQTL is based on robustly expressed genes (> 50 reads) and transcripts (FPKM >0.5) as mentioned on page 14 at lines 590-593. To enable power to detect variation in correlation strength by allele we selected independent g/t QTLs with FDR < 1⁻⁵ and MAF > 0.039 with homozygous minor allele counts of > 5 (Page 22, lines 866-867). It is important to note that our approach does not make assumptions as to neutrality or otherwise but is based on a statistically significant change in correlation between the cis gQTL and multiple genes in an allele specific manner. Crucially, our described coExQTL are replicated in an independent dataset. With reference to the example plots in Figure 5 where we analyse 3 coExQTL across the three test conditions, a pathway-based differential co-expression analysis was employed to determine whether the coExQTL disrupts the relationship with a pathway based on curated gene sets from pathway databases. The statistical approach used for ascertaining significance of the results for these coExQTL is described in the method sections have been updated with a new section to review the related results and methodology (page 23 lines 926). In addition, we have adjusted Figure 6 (page 51) to ensure we only include pathways with statistical significance. We provide all results for these figures in Supplementary Table 7 for all coExQTL as well as the three example pathway analyses.

9. *Section 2.2 lacks clarity on the application of conditional analysis for the analysis of tQTL. A more detailed explanation of the conditioning process within the analyses is*

necessary.

Author response:

Peak-eQTL are defined as those that remain significant association between a g/tSNP and a quantitative trait after conditioning on other SNPs within the same genomic region. Cis conditional pass analysis was utilized to detect peak-eQTLs. These explanations were included on pages 3 (lines 106-108) and the method section (pages 16, line 648).

10. Given the minor allele frequency (MAF) threshold of 0.04 and the sample sizes (n=174/139), the study may be underpowered for detecting effects associated with many of the included SNPs. A power analysis at the selected false discovery rate (FDR) level is recommended, with subsequent exclusion of potentially low-powered QTL.

Author response:

We added a new section (1.2) to the method and thoroughly reviewed the sample size calculation for bulk tissue eQTL analysis (page 13, line 544).

We excluded potentially low-powered eQTLs for downstream analysis using various strategies. We chose peak gQTLs ($FDR < 1^{-5}$, $MAF > 0.039$) with a homozygous minor allele count of greater than 5, in order to detect coExQTLs. We used causal variants (coloc posterior probabilities of $H4 > 0.8$) that were linked to both expression and methylation for mQTL analysis and used a threshold for FDR ($< 1^{-3}$) that was less liberal (Page 22, lines 866-867). The high degree of replication with the earlier array-based analysis further demonstrates the dataset is appropriately powered for this MAF threshold.

11. The method utilized for calculating the FDR is not specified. Given that the multiple p-values corrected for FDR are likely correlated (e.g., due to local CpG density affecting methylation, linkage disequilibrium among SNPs, interconnected genes within pathways, etc), an FDR calculation method that accounts for these dependencies is necessary.

Author response:

A new section (2.3) was added to the method, and we thoroughly reviewed the correction for multiple testing in eQTL analysis on page 17 (line 696).

12. There is a concern regarding unaccounted confounders or model misspecification leading to p-value inflation. The authors could estimate this inflation by presenting QTL p-value distributions as quantile-quantile (QQ) plots and calculating genomic inflation (lambda) values. Should inflation be detected, a re-evaluation of the models is warranted.

Author response:

We appreciate the reviewer's comments here, although note that quantile-quantile (QQ) plots are of limited utility for eQTL given the multitude of tests across a study (thus would be required for each gene/ transcript). Moreover, the nature of genomic structure means that the independence of association normally observed in QQ plots does not apply for cis eQTL which are all within the window of a gene and will have varying degrees of linkage disequilibrium.

We attempted to establish stable genome-wide significance thresholds for eQTLs and increase our confidence in our findings by taking into account the number of tests performed and using appropriate multiple testing correction methods (Page 17, line 696). We also included dominant PCs as independent variables in the regression model for g/tQTL analysis to regress out technical noise, batch effects, or other confounding factors (Page 15, lines 608-629).

13. The assertion that a relationship between cg22687766 methylation and CD55 gene expression is established post-LPS is premature without causality analysis or specific supporting data. Figures 3d and e, as well as Supplementary Figure 7, should include causality analysis or additional data to substantiate the proposed interaction between DNA methylation and gene expression. As it stands now, the relationship between methylation and CD55 expression can be explained by rs2914937 genotype that affects both, thus being a confounder.

Author response:

We thank the reviewer for this point. We agree that this is a complex association –our analysis suggests this locus is causal to mQTL formation both pre and post LPS, whereas gQTL/tQTL only form post-LPS. Thus, LPS is causal to the formation of the regulatory effect on expression, although we agree that LPS is not causal to the mQTL. Nonetheless, the regulatory locus for both effects appears to co-localise as shown by a posterior probability H4 value of 1. We have added this analysis (page 7, lines 262-265) and a descriptor paragraph in the revised manuscript. In addition, we expanded the explanations for causal analysis in the method section on page 21 (line 822) and the supplementary figure 7 (page 33, lines 1349-1350).

14. The manuscript would be more accessible to readers if it included schematic diagrams to illustrate the study design and concepts.

Author response:

We thank the reviewer for this comment – in acknowledgement of their advice we have added a new figure (Figure 1 on page 46) to provide an overview of the steps involved in performing eQTL analysis on monocytes.

15. There are inconsistencies in terminology and acronym usage throughout the manuscript. For example, the acronym "PP.H4" is not defined, and there are variations in the terms "QTL" vs. "QTLs," "IFN- γ " vs. "IFN γ ," and in 2.4.1 an error in "casual" instead of "causal".

Author response:

We thank the reviewer for their comment and apologise for the oversight. We made changes to the inconsistent terminologies throughout the text and corrected the typographic errors.

A new section has been added to the method (page 21, line 822) to define PPH4. In addition, we define PPH4 in the abstract (page 2 line 41) and results sections (e.g., page 6, line 235-239).

16. The authors should complete the legend for Supplementary Figure 1 by providing descriptions for panels a) and b). Furthermore, an explanation of the purpose and selection criteria for the 0-50 PCs tested in gene expression profiles is necessary.

Author response:

A new paragraph has been added to the method section to fully explain our objective of selecting and utilizing the optimal number of PCs to use as covariates in eQTL analysis (Page 15, lines 608-629).

Furthermore, we modified the description of Supplementary Figure 1 on page 53.

17. The font size in the main figures is occasionally too small for comfortable reading, and the contrast of the text in panel strips is too low to be readable. Figure 4 resolution is significantly lower than in others. To enhance readability, the font size in the main figures should be increased to at least 5pt.

Author response:

We remade and enhanced main and supplementary figures and raised the font size.

Reviewer #2:

Nassiri et al. have conducted an extensive bulk QTL study in immune-stimulated monocytes of 185 European donors. More specifically, they assessed methylation QTLs, both transcript- and gene-level eQTLs, and co-expression QTLs. Using linear regression and co-localization analysis, they found that many ($\geq 86\%$) epigenetic and expression-level changes share a likely causal variant. Finally, they co-localized the context-specific transcript- and gene-level eQTLs with GWAS risk loci and noticed that depending on the context different diseases are co-localizing with the eQTL signal.

While the analyses have been conducted with care, currently, the novelty of the study is more limited. Many context-specific eQTL (e.g. Fairfax et al. 2014; Kimm-Hellmuth et al. 2017) and (less so) co-expression QTL analyses (e.g. Oelen et al. 2022) have already been conducted in the past – also in monocytes, and by the authors themselves. Moreover, many of these efforts have been previously uniformly processed and combined in the eQTL catalogue, a compendium of gene expression, exon expression, transcript usage and splicing QTLs (Kerimov et al. 2021). The main novelty could have been combining the transcript-level eQTLs with the co-expression QTL analyses, however, this analysis was not conducted.

Major comments:

1. To increase the novelty of this study and make the manuscript more of a complete story, it would be very interesting to conduct the co-expression QTL analyses using the transcript eQTLs instead of the gene eQTLs. Currently, the manuscript feels like a lot of independent analyses have been put together without a clear storyline that connects them, this would be one way to make it a more coherent/connected story. Previously, it was shown that genetic variation can decouple/affect the upstream regulatory network of eQTL genes – at the gene eQTL level (e.g. van der Wijst et al. 2018; Oelen et al. 2022). By now conducting co-expression QTL analyses using the transcript eQTLs, you may find how different isoforms have different upstream regulators that may also provide an explanation for some of the isoform switching that you observed (e.g. OAS1, Fig. 2e).

Author response:

We thank the reviewer for this point and agree that the transcript co-expression is an interesting question to explore. Taking account their advice we have greatly developed our analysis and expanded the manuscript using the transcript eQTLs to explore the specific relationship between transcripts and other genes networks in an allele specific manner. We incorporate these analyses in a highly revised version of the manuscript which includes new figures including one dedicated to tQTL co-expression analysis (Figure 7, pages 52), supplementary table 9, and a new section called 'The genetic determinant of transcripts regulatory network relationships' on page 10.

2. Should the focus of the manuscript not be completely revised? Isn't the main message that tQTLs are much more informative than gQTLs? As they are more

enriched for disease associations, may overlap better with mQTLs (described in methods section that this was done, but it's not discussed in the results section at all – why?), novelty in transcript QTL co-expression QTL analysis, etc?

Author response:

We thank the reviewer for this point. As per our above comment, we have very heavily revised the manuscript to emphasize the importance and benefits of tQTL as well as rerunning our overall eQTL analysis using a conditional approach.

We demonstrate that using transcripts in addition to gene-level coExQTL yields a highly precise and biologically relevant understanding of gene regulation and its effects on complex traits. We further investigated genetic variants that were associated with both gene expression and DNA methylation in co-expression modules to acquire a more comprehensive comprehension of the regulatory mechanisms (e.g., page 8, lines 326-336 and Figure 5 on page 51).

3. The paper would greatly benefit from copy editing, as several sentences are way too long or are sloppy due to missing words or just generally very difficult to comprehend. As a result, the sentences are sometimes not understandable at all. To give a few examples of sentences that are way too long, miss words or that I did not understand at all:

a. 'Correspondingly, a significant proportion of monocyte eQTLs? are only apparent upon monocyte activation 3, 6.'

b. 'Further exemplar tQTLs demonstrating high degrees of conditional specificity include to the genes RGS1, DDX1, CTSC and KIFC3 (Supplementary Figures 4 and 5), as well as tQTL to genes where the same SNP influences tQTL formation in opposing directions to different transcripts according to immune state for example, MICA (Supplementary Figure 6).'

c. In untreated monocytes, this locus demonstrates weak gQTL activity (REPORT EQTL Z-SCORE INSTEAD?) (OAS1 FDR=0.0027, OAS3 FDR=0.0023), however expression of both these genes is robustly upregulated by exposure to IFN γ and this state correspondingly associated markedly increased significance of these associations (OAS1 FDR = 10⁻¹⁴; OAS3 FDR = 3⁻²¹) (Figure 2d).

d. One example of such an effect where context leads to establishment of a genetic relationship is at rs2914937 (chr1:207315423G>A) which is an mQTL for cg22687766 (PP.H4 = 1) within the upstream promoter (FDR 10⁻⁵⁸) across both resting and LPS treated states, and specific to post LPS state forms an gQTL for CD55 (FDR 10⁻⁶³), and an ??? (Figure 3c,d).

e. Genetic determinants of condition-specific expressed transcripts. This paragraph seems to be in a random position in the supplementary information. This part especially was badly written, and I had no clue what the entire paragraph was doing here.

Author response:

We are grateful to the reviewer for their frank appraisal and have worked to take into account these helpful comments as well as the previous valid point regarding focus of the manuscript. In doing so, the manuscript has been heavily rewritten, with a more focused narrative and better use of language.

4. To further enhance the coherence of the story: When you discuss the gQTL and tQTL results (in the 'Identification of cis-acting QTL' section), can you also mention something about the overlap between gene- and transcript-level results? So how many gQTLs are also a tQTL, how often are they concordant, how often are they condition-specific, etc?

We found that 54.6% of tQTL (4763 observed over 8727 total), showed context specificity, of which 3.4% involve more than one transcript associating with the same regulatory variant. ◊ e.g. are these tQTLs with one variant regulating multiple transcripts from the same gene more often a gQTL?

These types of analyses will give the reader a bit more information about how relevant it is to also look for eQTLs at the transcript-level instead of the gene-level (the level at which most studies are currently being conducted) and will give some more molecular insights into the underlying processes.

Author response:

We agree with the reviewer and have added the following paragraphs to page 4 (lines 160-165) in the revised manuscript to review the overlap between gQTL and tQTL.

"We found that 54.6% of tQTL (4763 observed over 8727 total), showed context specificity, of which 6.5% (568 observed over 8727 total) involve more than one transcript associating with the same regulatory variant (387/568 demonstrating context-specificity). Whilst 39.3% (4276 observed over 10875 total) of gQTL genes had a tQTL, with 16.6% (710 observed over 4276 total) of them being context-specific gQTL genes, notably 25.6% (1473 out of 5749) of tQTL were to genes without gQTLs. Thus, tQTL analysis provides additional information regarding context-specific regulatory variant activity that complements gQTL analysis."

5. To further enhance the coherence of the story: Why not integrate the DNA methylation information for the interpretation of the co-eqtls? Does it explain the context-specificity?

Author response:

We thank the reviewer for this point and in response have carried out a new set of analyses, including the coExQTL analysis that utilizes both gene expression and methylation data. This has complemented the previous coExQTL analysis and generated new data (e.g., page 8 at lines 326-336). The results were utilized to explain the specificity of gene coExQTL (Page 51, Figure 6d, h). We provide the full output of these analyses in Supplementary table 8.

6. Using a complementary Bayesian approach (Moloc)^{19, 20}, we identified evidence ($PP > 0.5$) supporting 3572 gQTL and 2016 tQTL being unique to one condition only (Supplementary Table 2). Could you provide more clearly what the complementarity is here? First, it is not made explicit to what exactly it is complementary. Second, it is not made very explicit what you assess with Moloc. Could you elaborate a bit more so that it is also clear for someone who doesn't have heard from Moloc before?

Author response:

The revised manuscript now includes text explaining the Moloc approach and how it can enhance the context-specific validity of g/tQTL findings as follows (Page 3, lines 114-118).

“The Moloc method was utilized to further evaluate the context specificity of g/tQTL identified in RNA-seq data enabling comparison of evidence for shared or independent effects of genetic variants whilst mitigating the impact of linkage disequilibrium. This approach identified 3572 gQTL and 2016 tQTL with evidence of specificity to one condition only ($PP > 0.5$, Supplementary Table 2).”

7. Using Mendelian randomization, we found that tQTLs are enriched for loci overlapping disease processes ($FDR < 1-10$ and $PP.H4 > 0.8$), with tQTL from the untreated monocytes showing marked enrichment for rheumatoid arthritis and cancer, and LPS- and IFN γ -stimulated monocytes showing enrichment for asthma (Figure 2a). ‘disease processes’ = GWAS for disease? Enriched compared to what? And more enriched than gQTLs? In other words, is it important to look at tQTLs in addition to gQTLs? Or what is the message that you want to convey here?

Author response:

On page 5 (lines 181-184), we included a new paragraph to examine the Mendelian randomisation (MR) test and expanded the related section in method (page 19 at lines 77-782). In the updated manuscript, we introduce MR-eQTL as a procedure that uses SNPs as instrumental variables to analyze the connection between gene expression and a particular outcome (i.e., GWAS traits).

On page 4 at lines 160-165 and page 5 at lines 166-167, we assessed the significance of examining tQTLs in addition to gQTLs.

8. For certain traits, disease enrichment using Mendelian randomisation (MR) was most significant for gQTL post treatment, such as MS and LPS associated gQTL, whereas for inflammatory bowel disease and COVID19 risk-loci, enrichment was most marked in IFN γ induced gQTL (Figure 2a-b and Supplementary Table 3). This part that follows is suddenly about gQTLs again. There is no connection in the storyline at all, it feels very disconnected. As mentioned in the comment above, by discussing the results in comparison (tQTL vs gQTL) you create much more of a connected storyline. Dont you also wanna explain whether these findings are as expected or not? So are

the diseases you mention known to be IFN-induced or association LPS and MS... Does that make sense?

Author response:

We thank the reviewer for this point and have revised the interpretation of the association with disease section correspondingly on page 5 at line 178.

9. *The reproducibility of the cis-gQTLs has been assessed against micro-array-based results from an earlier cohort. It would be also very interesting to see how many of the cis-gQTLs can be reproduced using pathogen-stimulated monocytes in scRNA-seq data (Oelen et al. 2022). Although the stimulations are slightly different in this study, they are for the same duration (24h) and many of the same biological processes will be activated by these pathogens. An important difference is that this study uses pathogen-stimulated PBMCs instead of isolated monocytes. So potentially some biological processes will not be activated in monocytes that are stimulated without the other immune cells present. Including such a comparison gives valuable, more general insights into the reproducibility between single-cell and bulk QTL studies, and which QTLs might be missed in an (even) more artificial system of isolated monocytes stimulated with a single cytokine (IFN γ) or pathogen (LPS) component.*

Author response:

The revised version of the manuscript included a comparison study using published single-cell RNA-seq data (Oelen et al. 2022) on page 4 at lines 150-159 to validate our initial bulk RNA-seq gQTL analysis.

10. *Co-expression QTL mapping: although the methods section describes co-expression QTL analyses conducted in all three genotype groups, the results section only reports the analyses conducted between homozygous ref and alt groups. Moreover, these results are not placed into the context of other similar publications: in bulk, e.g. Zhernakova et al 2017 describes the interaction between genetic variants and pairs of genes in interaction analyses, and in single-cell, e.g. Wijst et al 2018 and Oelen et al 2022 have leveraged single-cell measurements to calculate correlations per individual. Importantly, bulk as opposed to single-cell co-expression QTL analyses have the risk to identify false correlations due to Simpsons paradox (Li et al 2023), meaning that in bulk measurements the actual direction of effect may be masked as co-expression correlations are measured across donors instead of per donor. Therefore, replication of co-expression QTLs in a single-cell study is preferred (e.g. Oelen et al. 2022) over the replication in the microarray study.*

Author response:

We thank the reviewer for this point and have added a new paragraph to emphasize that the co-expression QTL mapping covers all possible genotype classes (page 22 at lines 884-890). We are amongst a number of groups who have previously reported co-expression QTL (e.g. PMID 22446964). Clearly the development of much larger cell-

type specific datasets has greatly expanded the utility of this approach and observations.

In light of the reviewer's valid points we have greatly extended the discussion section to put our results in the context of other similar publications (Page 12 at lines 470-478) and citations to "Zhernakova et al 2017" and "Li et al 2023" were added.

Regarding replication, the aim of this work was to demonstrate the robustness of our described coExQTL, many of which, such as that observed at *ERAP2* post IFN- γ , are completely novel but very specific to one state (and likely cell-type) in our data. Moreover, our approach employs the application of secondary allele-specific geneset enrichment analysis to the results of coExQTL, identifying pathways associated with expression of one allele - implying disconnection of target genes from gene regulatory networks in a context-specific manner. This is novel and a strength of our data but requires large numbers of associations to be determined.

We agree that single-cell analyses can be helpful in attempting replication, but the low sequencing depth per cell is a significant challenge that can reduce accuracy of gene expression quantification, particularly for genes that have zero-inflated expression, leading to sparsity of genes detected and limiting replication of the pathway-based approach we employ. It is important to recognise that whilst Simpson's paradox can manifest in bulk data analyses – our presented data is of one cell subset (monocytes) and thus the cellular heterogeneity typically observed in bulk analyses (e.g. PBMCs) does not apply here, greatly reducing propensity for such false positives.

Given the identical approaches in cell isolation and treatment, the same local population (although different individuals) and the size of the previous microarray-based approach, it was deemed the most appropriate dataset for which to attempt replication of coExQTL and associated pathways. The results justify this decision, but we intend to explore scRNA-seq dataset in future analyses.

11. Using a conservative approach and only described coExQTLs that demonstrate the same directional allele-specific differential gene correlation analysis (FDR<0.01) for the same gene pair, SNP, and direction of correlation across both datasets. +We observed 1576 genes to correlate with ERAP2 expression in a genotype specific manner, of which 764 (48.4%) replicated in the same genotypically divergent direction, the most significant being EPM2AIP1 (Figure 4h). Could you report which proportion did not replicate in the micro-array study? Could you report the concordance (same direction of effect) of all significant effects from the bulk RNA-seq study, in the micro-array study? Again, replication in a single-cell study is preferred to overcome the limitations of bulk studies for co-expression QTL mapping.

Author response:

We revised the related section on page 7 at lines 297-310. We elaborated on the direction of effect and examined the challenges that caused our failure to duplicate coExQTL from the bulk RNA-seq study in the microarray study.

Unfortunately, we were unable to find a coExQTL based on scRNA-seq that could be fully compared to our data. We had the thought of performing coExQTL analysis ourselves on published scRNA-seq profiles with raw data. The process of accessing the raw data has been a complicated one, and we are still in the middle of the paperwork. On the other hand, scRNA-seq profiles have a limited sample size, which makes replication less accurate.

12. Pathway analysis of co-expression QTL genes: did you use a background gene set, and if so, which one?

In the methods section you describe a differential gene expression analysis but it is not clear where in the main text this is described.

Author response:

The hc2.all.v2024 reference gene set that was used for pathway-based differential co-expression test was reviewed on page 7 line 289 and page 24 at lines 959-963. The gene set hc2.all.v2024 is populated by a diverse array of pathways that encompass a broad range of biological processes.

Although we weren't doing 'differential expression analysis' specifically, we utilized certain data preprocessing steps for RNA-seq differential expression analysis, such as variance stabilizing transformation. The objective was to take into account the differences in library size and other technical factors that could affect gene expression measurements. We made changes to the method section to make it clear (Page 14 at lines 584-586 and page 15 at lines 587-588).

13. Despite many efforts to construct personalized and context-specific gene regulatory networks^{18, 47, 48}, there has not been a systematic effort to apply differential gene correlation analysis in an allele-specific manner. This is incorrect: see for example , Wijst et al. 2018, Oelen et al. 2022, Li et al. 2023.

Author response:

We thank the reviewer for this point and have appropriately cited these papers. We have revised our language, the emphasis now being that transcript-level expression profiles were not utilized in the previous published coExQTL analysis (Page 12 at lines 470-478).

Additionally, we expanded the discussion section to highlight the importance of integrating multiple layers of genomic information to uncover complex regulatory mechanisms (Page 12 at lines 487-493).

14. *There is already quite some literature on the role of OAS1 in COVID-19 (e.g. and <https://www.ncbi.nlm.nih.gov/pmc/articles/PMC8288155/>), but the findings ('notable disease relevant associations (PP.H4>0.99) include IFN γ specific tQTL at COVID-19 severity locus rs10735079, where allelic variation modulates context-specific splicing of OAS1, ...') are not really discussed in light of what has been already reported.*

Author response:

A new paragraph was added on pages 6 (lines 213-226) and our findings were compared to previously published results.

Minor comments:

1. *Can you briefly explain in the results section why you have chosen different windows for the gene vs transcript eQTLs? This helps the reader to better understand why things are being done the way they are presented.*

Author response:

We added a new paragraph to the section results to examine the reasoning behind choosing different windows for the gene versus transcript eQTLs (page 3, lines 101-105). In addition, multiple citations were included to back up the use of the 100kb window for tQTLs (page 3, line 122 and page 4 at lines 123-128).

2. *Throughout the manuscript you use gQTL, tQTL and eQTL. You would expect that eQTL refers to both gQTL and tQTL here, but it seems to only refer to gQTL. Shouldn't you just keep using gQTL and tQTL throughout the manuscript to prevent any confusion?*

Author response:

The reviewer raises an important point in readability - the article commences with the use of the term 'eQTL', which is both representative and generic. The terms gQTL and tQTL are defined on page 2, lines 68-72. As suggested by the reviewer, we replaced the eQTL with gQTL or tQTL after that part to avoid confusion.

3. *It is unclear whether this difference in window size may have impacted your results?*
a. *Whereas the median distance for peak gQTL from the transcription start site (TSS) was 14648 bp (95% CI [13982 – 15435]), peak tQTL are typically more proximal to TSS (median 5822 bp 95% CI [4938 – 6598] (UT), 6350 bp 95% CI [5176 – 7362] (LPS), 6248 bp 95% CI [4901 – 7688] (IFN γ)).*

15. *Did you limit your analyses only to the 100 kB window size for both the tQTLs and gQTLs? If not, you should confirm that the observed differences between tQTLs and gQTLs are not a consequence of the different window sizes (1Mb for gQTL vs 100 kB for tQTL) that were used for the discovery.*

Author response:

We examined if despite the impact of window size on the identified gQTL and tQTL associations, it is still possible to make significant comparisons between the two types of eQTL (page 3, line 122 and page 4 at lines 123-128).

16. For the interpretation / comparison of tQTL and gQTL results in Figure 1D it is relevant to mention in the legend that a different mapping window is used, as this may affect the relative distribution to which genomic feature a QTL is assigned to. So does this imply that gQTL and tQTL cannot be fairly compared in Figure 1D and 1E and should be presented in separate figures? Or does it mean that any conclusions drawn from here that are cross gQTL and tQTL should be confirmed in a 100 MB window gQTL mapping? 4. Fig. 1D-1E: Similarly, the number of gQTL and tQTL within a specific genomic feature or chromatin state may vary, even independent of the chosen window size. As a result, the gQTL and tQTL cannot be fairly compared within the same feature or state. So does this again imply that gQTL and tQTL cannot be fairly compared in Figure 1D and 1E and should be presented in separate figures?

Author response:

We analyzed whether it is feasible to compare gQTL and tQTL enrichment with GWAS trait susceptibility, despite the impact of window size on the identified associations. By conducting an analysis with the 100Kb window size for peak gQTLs, we were able to confirm our earlier conclusions. The 90.9% GWAS trait was able to be replicated (Page 5, lines 192-196). Additionally, we included the results as new sheets in supplementary table 4.

In addition, we point out that gQTL and tQTL are mapped differently in the legend of figure 2d (figure 1d in the previous version of the manuscript) (Page 6, lines 13-16).

5. Please write out abbreviations in full the first time they are being used, e.g. 'TxFlnk marks'.

Author response:

We thank the reviewer for this comment. We have now defined the abbreviations in the text (page 3, line 134), in the legend of figure 2 (page 27, lines 1078-1084), and table 2 was created in the method section to give a full description of each (page 20).

6. All p-values/numbers in the manuscript (main text and figures) are weirdly presented and don't seem to reflect the actual accuracy of the presented value: e.g. not 1-14 but 1.00×10^{-13}

Author response:

We apologise for these inconsistencies and have revised all p-values in the main text and figures as per the reviewer's instructions.

7. For all figures: when the QTL effect is not significant, 'na' is reported. Could you give

the actual p-value instead, and highlight significance for example using asterisks? Additionally, a rho, Z-score or slope should be given for all the QTL plots.

Author response:

Many thanks for this point. We have replaced all the “na” values in the figures with actual p-values and highlighted the significance using asterisks or a different background for the state name. We also added slope values to all the QTL plots.

8. For all figures: could you mention in the figure legend what the error bars represent and the N for each figure is? Could you also mention in which measurement the Y-axis is displayed (Fig. 1F: e.g. ‘Gene expression’ in FPKM?)

Author response:

We included the “N” for every boxplot to specify the number of donors. Furthermore, we examined a summary of five numbers for a boxplot in the figure legend and mentioned which measurement is displayed on the Y-axis.

9. For all figures: readability should be improved by increasing the font size of all axes and numbers displayed.

Author response:

To enhance the resolution and font size, we regenerated the main and supplementary figures.

10. rs4072037 allele carriage ◊ unclear: you should specify which SNP allele you mean (the risk allele apparently after reading the rest of the sentence, so which allele is that?)

Author response:

On page 4 (lines 171-172), we included the potential carriage alleles and risk alleles for rs4072037.

11. but which notably had no gQTL post IFN γ 24 (Figure 1g). ◊ do you mean that it doesn’t have a gQTL but did have a tQTL? Or do you mean it has no gQTL after IFN γ stimulation but did have a gQTL in unstimulated? The way you write it, it seems to imply the latter, but I think you mean the first. Better to rewrite to make this more clear, e.g. ‘We found IFN γ exposure induced tQTL (but not gQTL) ...’

Author response:

The sentence was revised on page 5 (lines 168-173).

12. Fig. 1A-1C: it’s unclear why specific eQTL examples are highlighted. Would be good to clarify this in the figure legend.

Author response:

On page 27, we improved the figure 2 legend by adding the following sentence (page 27, lines 1062-1063):

“Top condition-specific independent SNPs can be found in highlighted SNPs on Manhattan plots.”

13. *Fig. 1D: (d) Functional consequences of lead tSNPs on genomic features.* ◇ *‘functional consequence’ should be rewritten, e.g. as ‘functional annotation of the genomic feature wherein the lead tSNP resides’, as you don’t assess the consequence of the SNP but just annotate the genomic feature in which this SNP is located, right? In comparison with background SNP.* ◇ *How is the background SNP defined here?*

Author response:

The text now uses 'functional annotation' instead of 'functional consequence'. Furthermore, we added the sentence below to the caption of figure 2 on page 27 (lines 1068-1071).

“The foreground SNP sets for each gene/transcript were made up of g/tSNP with FDR under 0.001, and other SNPs within the 1Mb window around the TSS were used as background.”

14. *Fig. 1E: peripheral blood (E029)* ◇ *unclear what this means or refers to.*

Author response:

The explanations for E029 have been included in the text (page 19, lines 787-788 and page 27 at lines 1071-1072).

15. *Fig. 1F: The arrows in fig. 1F are distracting from the point you want to make as they are missing in some introns and don’t highlight the actual differences among the transcript variants. It’s maybe also good to highlight where the differences in transcript isoforms are in the two highlighted isoforms? Because apparently that’s the region that explains the different downstream consequences! Currently, it’s almost like a find-the-differences puzzle.*

Author response:

A new approach has been used to visualize and annotation alternative isoforms in the revised version of the manuscript. The new method of visualization was utilized in figures 2 (page 47), 3 (page 48), and 7 (page 52). Moreover, because boxplots also represent expression levels, we chose to eliminate extra plots for displaying the level of transcript and gene expression.

16. The abbreviations that are used in Fig. 1E and 1F are unclear and therefore should be explained in the figure legends.

Author response:

The abbreviations were explained in the figure legend (page 27, lines 1078-1085) and a new table was added to the method section to fully describe them (Table 2 on page 20).

17. Fig. 2B: The figure seems to imply that disease co-localization of tQTLs is better than for gQTLs, and that the naive condition shows highest co-localization. However, the amount of gQTLs and tQTLs differ already widely per condition, so shouldn't it be more informative to show the proportion of QTLs that co-localize as well? Additionally, to show that the context-specificity is so important for capturing disease-associated effects, could you also show an extra bar with those tQTL or gQTL that are independent of condition?

Author response:

A new section was added to figure 2 (Figure 3b in the revised manuscript – page 48) that displays the proportion of eQTLs that co-localize with GWAS traits. Moreover, we added additional bars to cover tQTL or gQTL that are independent of the condition.

On page 5 at lines 185-191, the advantage of employing tQTL in GWAS enrichment analysis is highlighted.

18. Fig. 1F and 2C: It's confusing that there is more isoforms with IFNg vs LPS (in isoform usage figures) - why is that? If there's neglectable expression, it may still be good to keep all isoforms in there, so it's clear? And why does this even need 2 separate figures (both 1F and 2C)? Isn't it very easy to put all information in one figure? Now a lot of data is repeated twice (UT) and it doesn't make it easier to compare when they are in different plots. Fig 2C  x-axis label 'LPS24' seems incorrect. Y-axis label can be more clear  why not specify that it's 'OAS1 expression level' or 'OAS1 isoform expression level'

Author response:

A new approach has been used to visualize and annotation alternative isoforms in the revised version of the manuscript. The new method of visualization was utilized in figures 2 (page 47), 3 (page 48), and 7 (page 52). Moreover, because boxplots also represent expression levels, we chose to eliminate extra plots for displaying the level of transcript and gene expression.

19. Fig. 2D-2F: the figures are not logically ordered. Even though the same SNP is affecting both OAS1 and OAS3 gQTL effects, the OAS3 gene does not come back in Fig. 2C or 2E, so it's distracting/confusing. I would suggest to 1. remove OAS3 and

IFNAR2 all together, or 2. to group OAS1, OAS3 and IFNAR2 gQTLs as one figure being the 'COVID19 risk loci associated gQTLs', and position it before fig. 2C. Old Fig. 2C and 2E can then follow as an example of such a risk locus that is further explained by tQTL effects, but maybe it's good to then highlight that no tQTLs were observed for OAS3?

Author response:

We deleted OAS3 from Figure 2 (figure 3 in the revised version – page 48) in the revised manuscript and rearranged the sections.

20. Figure 2C and 4J are not mentioned in the main text. Please make sure that all figures are referenced and discussed, in the order they appear in the figures.

Author response:

We have revised all figure numbering and ensure all are referenced and discussed in the order that they appear in the figures.

21. The differential splicing of ENST00000452357 skips an exon in ENST00000202917 with rs10735079 (chr12:112942203G>A) ∅ it would be good to highlight in Fig. 2C which exon is skipped, so that the figure is more informative and directive for the reader where to look at.

Author response:

On page 48, the skipped exons in figure 2c (figure 3c in the revised manuscript) were highlighted and revised.

22. Fig. 3A: Figure looks beautiful, but I'm not sure what message you want to bring with this figure? Just a look-up table might be way more practical - and if the message isn't clear from this figure anyways then maybe removing it all together is better (or improving the text and legend to make the message clear)?

Author response:

The purpose of this figure is to present the 100 LPS-specific g/tQTL-mQTL pairs that have a causal variant that affects both gene/transcript expression and methylation level. The text and legend were improved to make the message clear (Figure legend: page 28 at lines 1132-1136 and page 29 at lines 1137-1138 - Main text: page 7 at lines 254-256).

23. Fig. 3B: again maybe not the best way of visualizing the main message. Don't you want to know the relative proportion of each category within the positive vs negative effects (so scale differently)?

Author response:

The figure was modified by including the proportion (figure 4 in the revised manuscript on page 49) and expanding the related explanations in the figure legend (page 28 at lines 1121-1130).

24. *The departure from independence is coloured according to whether the residuals are positive or large. ♦ 'or large': no clue what you mean by this?*

Author response:

This sentence has been removed now that we have revised the figure legend (figure 4b in the revised manuscript on page 49; figure legend on page 28 at lines 1121-1130).

25. *Fig. 3E: The adjusted p-value for the beta coefficient correlation between methylation and expression is 4.56×10^{-6} . 17. This p-value is independent of the genotype, or was it written unclearly in the legend? Additionally, could you explicitly test for a genotype x condition (unstim vs LPS) interaction effect instead, and report that p-value in Fig. 3E? And don't forget to report the rho of the regression lines and the non-significant p-values.*

Author response:

The adjusted p-value for the beta coefficient correlation between methylation and expression independent of the genotype is 4.56×10^{-6} . The figure now includes the p-value for the beta coefficient correlation between methylation and expression for naïve as well (page 50).

It's important to point out that this isn't an example of coExQTL between gene expression and methylation. To avoid confusion, we avoid including details similar to coExQTL. The coExQTL between expression and methylation will be discussed in the next sections (pages 7 and 10), where we will provide detailed information about the method and examples.

26. *We confined our analysis to lead-gQTLs with $FDR < 10^{-5}$ and $MAF > 0.039$. 18. Can you explain in the text why you have made these decisions (the numbers seem arbitrary at first glance)?*

Author response:

In order to balance sensitivity and specificity, we limited our analysis to peak-gQTLs with $FDR < 10^{-5}$ and $MAF > 0.039$. This was explained on the page 16 at lines 682-694.

27. *Correlation coefficients (r) were computed for the expression values of eGenes and all other genes for individuals homozygous for either allele... 19. Can you also specify in the main text that this is the Spearman r?*

Author response:

We added this on line 148 of page 4.

28. *Supplementary tables: can you provide an explanation for all the column names in the first tab of each table?*

Author response:

The first tab of each supplementary table now has an explanation for all column names.

29. *An allele that shows recent selective pressure, purportedly due to the Black Death³⁶ 20. There has been quite some commotion in the field about the paper that is being cited here, see for example <https://www.ncbi.nlm.nih.gov/pmc/articles/PMC10055098/>. Please be careful with the specific citation, maybe better to remove this remark all together as the scientific community is not convinced about the statements made in the cited ref 36.*

Author response:

We thank the reviewer for alerting us to this and given the controversy have removed this section from the revised manuscript.

30. *Fig. 4B, 4E, 4H: You report on the adjusted p-value of the correlation difference between the two stimulation conditions (naïve vs LPS). Can you better explain what exactly has been tested? Did you put the stimulation condition as interaction term?*

Author response:

To find genetic variants that affect gene co-expression patterns (coExQTL), we create gene co-expression networks for every genotype-matched group. Next, Spearman's rank correlation coefficient (r) is computed for the expression values of eGenes and all other genes among individuals stratified according to genotype. We use the correlation coefficient as edge weights and assess whether the differences in edge weights between genotype groups are statistically significant (see method for more details). We expanded the related explanation in the revised manuscript on pages 7 (lines 281-289) and page 22 (lines 884-890).

31. *Fig. 4I: unclear what the red vs blue color represents.*

Author response:

Apologies for confusion here – in light of the reviewer's comments this figure has been recreated in the revised manuscript with a uniform colour scheme (figures 6 in the revised manuscript – page 51).

32. Post LPS, coExQTLs are present at rs3110426 and OXR1 expression, with rs3110426 possibly affecting the binding site of ZNF628 transcription activator³⁸.
◇ Unclear how this reduced binding would then result in a negative co-expression relationship (instead of the absence of any co-expression relationship).

Author response:

This section has been removed from the revised manuscript.

33. *Whilst we believe this work has furthered our understanding of the impact of immune conditions on regulatory genetics in monocytes, using LPS or IFN γ single treatment models inevitably will fail to recreate the complex cytokine milieu and cell-cell interactions that characterize the immune system in health and disease. 21. All these issues are largely mitigated using stimulated PBMC single-cell data (e.g. Oelen et al. 2022).*

Author response:

The reviewer makes an important point. Whilst PBMC stimulation assays can recapitulate cell:cell interactions, in itself this adds further variation to responses. Moreover, it would be wrong to assume stimulated PBMC assays - where cells are artificially fixed in concentrations markedly divergent to that in the blood or sites of immune reaction – form a particularly accurate model of *in vivo* responses. The aim of our approach was to maximise the discernment of genetic contribution to monocyte transcriptomic responses to two separate stimuli – introducing other cell types would only have increased the variance in the system. In acknowledgment of the reviewer’s point however we have added a reference to Oelen et al. 2022 and highlighting the importance of conducting similar studies at the transcript level (Page 2 at lines 64-72 and page 11 at lines 441-445).

34. *192 healthy individuals recruited via the Oxford biobank \diamond reference to Oxford Biobank is missing, and ancestry is not reported.*

Author response:

A citation was added to the Oxford Biobank and the ancestry of the recruited donors was mentioned (Page 13, lines 519-521).

35. *The shiny app browser works well and makes it easy for others to take advantage of the (significant) results. Also putting the non-significant eQTL summary stats somewhere would be appreciated by the community.*

Author response:

We are pleased that the reviewer was able to use the Shiny app and we’re hopeful that the community will find this of significant assistance. We thank the reviewer for their comment and have now provided non-significant eQTL summary statistics included.

https://livedataoxford.shinyapps.io/fairfaxlab_supplementary_files/

36. *Section 2.2 methods: ‘zero to 50 PCs ... ‘dominant PC’...’ ∅ what does dominant mean here, please elaborate?*

Author response:

On page 15 at lines 608-629, we have improved our explanation of how to determine the optimal number of PCs.

37. *Section 2.2.1 methods: Why did you not use coloc or fine-mapping for identification of the lead SNP?*

Author response:

Our approach to identifying “independent eSNPs” is based on conditional pass analysis, which is similar to fine-mapping. We rename the section 'Lead and independent SNP identification' (Page 16 at line 648) to review our approaches for identifying Lead (Page 16 at line 649-667) and independent eSNPs (Page 16, lines 668-673 and page 17 at lines 674-681). Furthermore, we bring up the application and benefits of each.

38. *Section 2.4.1 methods: Why did you not use Susie to directly incorporate LD information while colocalizing with GWAS signal instead of manually looking for LD proxy SNPs when the SNPs were not in the eQTL output itself already?*

Author response:

The TwoSampleMR workflow automatically searches for LD proxy SNPs if they are not in the eQTL output (<https://mrcieu.github.io/TwoSampleMR/index.html>).

We use the “proxies = TRUE” option in the “extract_outcome_data” function (https://mrcieu.github.io/TwoSampleMR/reference/extract_outcome_data.html). Next, we utilize colocalisation analysis to determine if PPH4 for colocalization of query SNP and its proxy exceeds 0.8. If the answer is affirmative, we utilize the Two-Sample MR statistical method to examine the causal connection between a genetic variant and a GWAS trait.

The development of TwoSampleMR is to search MR-Base database (<http://www.mrbase.org>). In order to remain consistent with common practice, we utilized the default approach (i.e., TwoSampleMR) and improved it by incorporating colocalisation analysis.

It is certain that incorporating more sophisticated approaches would improve this part.

39. *Fig. S2: the difference between S2A and S2B is unclear, no information is provided in the legend text. Figure S2C is very complicated. There is many t-statistics provided, and it's very much a puzzle to figure out what each of these represent. Please improve the figures and/or explain the figures better in the legend.*

Author response:

We improved the figure by replacing parts A and B and removing part C (Page 54). Furthermore, we significantly expanded the figure legend to clearly explain each part (Page 32, lines 1272-1285).

40. Multiple times throughout the figures/text 'LPS24' is mentioned, but it is unclear why it is not just LPS as in many other figures? It's always 24h LPS stimulation, right? So why not keep it consistent and refer to it as 'LPS'?

Author response:

LPS was used throughout the text in the revised manuscript.

Reviewer #2 (Remarks on code availability):

The code is not well organized and the description of required software, runtime, ordering and inputs of scripts needs to be greatly improved. The readme at the root of the repository only states the title of article with no other instructions supplied. There is little in the way of describing what analyses are found in which scripts and folders. The comments in the files themselves are also very scarce, giving little explanation to where specific steps are taken, and for which reasons. A great deal more detail would be required for someone to feasibly replicate the results described in the paper.

Author response:

We made sure to fully annotate the scripts, add explanations for each, and include the necessary software and runtime. The script can be found at the link below:
https://livedataoxford.shinyapps.io/fairfaxlab_supplementary_files/

Responses to Reviewer's Comments

We appreciate very much the reviewers for the constructive comments, and the effort and time put into the review of the manuscript.

Each comment has been carefully considered point by point and responded to. The reviewer and changes in the revised manuscript are as follows.

REVIEWER COMMENTS

Reviewer #2

We are happy to see that the authors have significantly improved the content and readability of the manuscript. The story has become much more coherent, and the novelty has very much improved. A few small remarks that have remained:

Specific Comments:

1. *The font size of the figures is still too small to read (at least in the merged pdf that was provided for review).*

Author response:

We raised the font size of images as much as we could and cut the margins of pdf files, which enhances their readability.

2. There is still a few textual errors in the main text, so a final proof-read would do good.

Author response:

We carefully reviewed the text and rectified the textual errors.

3. Some text is still too complicated / unreadable: e.g. 'Here IFN-exposure was found to induce tQTL (but not gQTL) to alternative transcripts of MUC1 (Figure 2f) including ENST00000612778 (FDR 2.7×10^{-6}) and ENST00000620103 (FDR 8.0×10^{-15}) according to rs4072037 carriage, a risk allele (rs4072037:G) for oesophageal and gastric cancer, illustrating the disease informing potential of this analysis (Figure 2g).'

Author response:

The text was carefully reviewed, and the complicated sentences were replaced.

4. In the naïve state, 36% (179/497) of gQTL and 42.4% (366/862) of tQTL demonstrate dependent causal effects on methylation linked to expression or vice versa, with this proportion being 40.3% (147/365) of gQTL and 43.3% (256/590) of tQTL post-LPS.

 Could you split this directionality up by methylation  expression or expression -> methylation? The same for the following sentence about positive vs negative relationship between methylation and expression.

Author response:

A new paragraph has been added to page 7 that details the statistics in Figure 4b.

5. It's difficult to assess whether the co-expression tQTLs are specific for one isoform, or entail potentially multiple isoforms from the same gene. Additionally, if different SNPs regulate different isoforms from the same gene, do they have similar or different upstream co-expression QTL genes that are in co-expression with these tQTL effects?

Author response:

The following paragraph has been added to page 10.

Context-specific transcript-level coExQTL analysis indicated that tSNPs specifically influencing the co-expression of a single transcript was the predominant mode of regulation in 98.4% of cases (1480/1507). In the remaining instances where different tSNPs regulated the co-expression of multiple transcripts from the same gene, we found that 34.7% (8/23) of these exhibited overlap in their upstream coExQTL genes, suggesting some shared regulatory mechanisms.

Reviewer #2 (Remarks on code availability):

The code and descriptions are much clearer now, thank you.